# Central tropical Pacific convection drives extreme high temperatures and surface melt on the Larsen C Ice Shelf, Antarctic Peninsula

Kyle R. Clem [1✉], Deniz Bozkurt [2,3,4], Daemon Kennett [1], John C. King [5] & John Turner[5]

Northern sections of the Larsen Ice Shelf, eastern Antarctic Peninsula (AP) have experienced dramatic break-up and collapse since the early 1990s due to strong summertime surface melt, linked to strengthened circumpolar westerly winds. Here we show that extreme summertime surface melt and record-high temperature events over the eastern AP and Larsen C Ice Shelf are triggered by deep convection in the central tropical Pacific (CPAC), which produces an elongated cyclonic anomaly across the South Pacific coupled with a strong high pressure anomaly over Drake Passage. Together these atmospheric circulation anomalies transport very warm and moist air to the southwest AP, often in the form of "atmospheric rivers", producing strong foehn warming and surface melt on the eastern AP and Larsen C Ice Shelf. Therefore, variability in CPAC convection, in addition to the circumpolar westerlies, is a key driver of AP surface mass balance and the occurrence of extreme high temperatures.

[1] School of Geography, Environment and Earth Sciences, Victoria University of Wellington, Wellington, New Zealand. [2] Department of Meteorology, University of Valparaíso, Valparaíso, Chile. [3] Center for Climate and Resilience Research (CR)2, Santiago, Chile. [4] Center for Oceanographic Research COPAS COASTAL, Universidad de Concepción, Concepción, Chile. [5] British Antarctic Survey, Natural Environment Research Council, Cambridge, UK. ✉email: kyle.clem@vuw.ac.nz

The Larsen Ice Shelf, located on the eastern side of the Antarctic Peninsula (AP) (Fig. 1a) has lost approximately 18,000 km² (20%) of its surface area since 1995[1–3]. The northernmost section, the Larsen A Ice Shelf, disintegrated in the summer of 1995[4], shortly followed by the collapse of the larger Larsen B Ice Shelf to its south in the summer of 2002[5,6]. South of

Larsen B lies the largest remaining section, the Larsen C Ice Shelf, which has been experiencing thinning[7] and dramatic rifting and calving in recent years, including a 5800 km² section that broke away in 2017 forming what was the planet's largest iceberg (A-68) at the time[3]. The loss of these ice shelves has caused thinning and acceleration of the tributary glaciers that fed the former ice

**Fig. 1 Study area and interannual climate variability governing Larsen C surface melt. a** Map of the AP region showing the northeast AP weather stations, ETOPO1 topography, locations of the former Larsen A and Larsen B Ice Shelves and the remaining Larsen C Ice Shelf, and the region used to calculate Larsen C surface melt (pink polygon). (left) The DJFM detrended correlations (shaded), 1991–2015, of Larsen C surface melt with **b** SST, **d** OLR, **f** 250 hPa streamfunction, **h** 500 hPa geopotential height. (right) The DJFM timeseries of (blue) Larsen C surface melt with (orange) **c** the SOI, **e** CPAC OLR anomalies, **g** Marshall (2003) SAM index, and **i** Drake Z500 anomalies. The bold black contours in (**b**, **d**, **f**, **h**) denote correlations statistically significant at $p < 0.10$. The CPAC OLR and Drake Z500 regions are denoted by green dashed boxes in (**d**) and (**h**), respectively. The linear trend lines of each time series in (**c**, **e**, **g**, **i**) are shown as a dashed line.

shelves[6,8,9], resulting in an increasing rate of contribution to global sea level rise; approximately 20% (2.5 mm) of Antarctica's total sea level rise contribution since 1979 has come from the AP[10], and mass loss from the AP has been accelerating and increased by around 15 Gt yr$^{-1}$ since 2000[11].

The loss of ice shelves and subsequent ice sheet retreat on the eastern AP is linked to anomalous warm summer surface air temperatures that result in strong surface melt events[12–14], which can lead to meltwater ponding and hydrofracture[15] due to the relatively thin snowpack on the Larsen Ice Shelf that allows the formation of very dense firn[16,17]. Indeed, the loss of the Larsen A and B sections were preceded by anomalously warm summer surface air temperatures and widespread surface melting[13,14], which led to the dramatic collapse of the Larsen B Ice Shelf over a period of just a few weeks[18].

These extreme high surface air temperatures on the eastern AP are commonly associated with foehn winds[19–21], during which air with a high potential temperature at high elevations of the AP (which has an elevation of over 2 km) warms strongly at the dry adiabatic lapse rate (around $+10\,°\text{C km}^{-1}$) as it descends the eastern slope of the AP onto the Larsen Ice Shelf. This process can be caused both thermodynamically by the slow cooling of air ascending the western slope of the AP at the saturated adiabatic lapse rate, or dynamically by mountain waves when airflow across the mountain barrier is blocked at low levels[22]. Foehn events can lead to localized regions of very high surface air temperature that can exceed $+10\,°\text{C}$[19], resulting in regions of strong surface melt on the Larsen Ice Shelf[20,23,24]. Indeed, the increased surface melting that triggered the collapse of Larsen A and B has been qualitatively linked to the concurrent strengthening of the summertime circumpolar westerly winds and the associated positive trend in the southern annular mode (SAM) index[25,26] primarily from stratospheric ozone depletion[27,28], which is suspected to have increased foehn events on the northeast AP[19,29]. However, the influence of the circumpolar westerlies/SAM on long-term variability and trends in surface melt across the Larsen Ice Shelf, especially the occurrence of extreme surface melt events that can lead to ice shelf collapse, has not been quantitatively demonstrated. Furthermore, more recent studies suggest strong surface melt on the remaining Larsen C section may be associated with remote forcing from the tropics[30].

Here we show that the atmospheric circulation pattern associated with extreme summertime foehn warming and surface melt on the Larsen C Ice Shelf is triggered by anomalous convection in the central tropical Pacific (CPAC) (10–15°S, 170–165°W). Summertime surface melt on Larsen C shows no significant relation to the SAM/circumpolar westerlies over our period of study (1991–2015), and is instead tied to a strong high-pressure anomaly over Drake Passage caused by the anomalous circulation pattern forced by CPAC convection. Therefore the SAM influence on surface air temperature and surface melt appears mainly confined to the northeast region of the AP[26,31], consistent with the lack of coherency in surface melt trends between the northeast Larsen Ice Shelf (i.e., north of the northernmost Larsen C embayments) and the Larsen C Ice Shelf in recent decades[13].

**Table 1 Larsen C surface melt seasonal relationships with regional and large-scale climate variability.**

|  | *MAM* | *JJA* | *SON* | *DJF* | *DJFM* |
|---|---|---|---|---|---|
| Niño 3.4 | **0.35** | 0.04 | −0.08 | 0.08 | 0.08 |
| SOI | −0.29 | −0.23 | −0.03 | −0.22 | −0.16 |
| SAM | **0.39** | <u>**0.43**</u> | 0.03 | 0.23 | 0.32 |
| CPAC OLR | −0.29 | −0.08 | 0.13 | <u>**−0.51**</u> | <u>**−0.63**</u> |
| Drake Z500 | 0.06 | 0.14 | 0.02 | <u>**0.67**</u> | <u>**0.65**</u> |

The seasonal detrended correlations, 1991–2015, of Larsen C surface melt with Niño 3.4 SST anomalies, the Southern Oscillation Index (SOI), the Marshall (2003) SAM index, CPAC OLR, and Drake Passage 500 hPa geopotential height (Drake Z500) (Methods). Correlations significant at $p < 0.10$ are boldface and at $p < 0.05$ are underlined.

Although the circumpolar westerly winds are projected to continue strengthening over the remainder of this century due to increasing greenhouse gases[32], these findings suggest variability in CPAC convection will be a key driver of the future stability of Larsen C and the remaining section of Larsen B, which will govern the AP's mass balance and its future contribution to global sea level rise.

## Results

**Larsen C surface melt relationship with large-scale climate variability.** Linear relationships between Larsen C surface melt (Methods) during the extended summer melt season (December-March, DJFM) with various climate parameters are shown in Fig. 1 and Table 1. Positive surface melt anomalies are broadly associated with an El Niño-like sea surface temperature (SST) anomaly pattern, with weak positive SST correlations across the central and eastern tropical Pacific (Fig. 1b). However, correlations are significant only in the southeast sub-tropical Pacific and in the southwest mid-latitude Pacific east of New Zealand, and the SST correlations over the equatorial Pacific are close to zero. Furthermore, there is no co-variability with the Southern Oscillation Index (SOI; Fig. 1c and Table 1). This suggests no significant relationship between Larsen C surface melt and the El Niño-Southern Oscillation (ENSO).

The positive SST correlations in the southeast sub-tropical Pacific lie in a region of strong climatological subsidence (Supplementary Fig. 1a) that strongly prohibits updraft formation, and positive SST anomalies here are unable to force a high-latitude Rossby wave response[33]. Therefore, it is unlikely these positive SST correlations have a physical connection to remote circulation anomalies governing Larsen C surface melt. However, correlations with tropical convection (Fig. 1d)—the physical mechanism required to generate a Rossby wave-show enhanced Larsen C surface melt is strongly and significantly associated with negative outgoing longwave radiation (OLR) anomalies (enhanced deep convection) in CPAC (Fig. 1d, e) with a strong detrended correlation of −0.63 ($p < 0.01$; Table 1). While the OLR correlations also broadly reflect an El Niño pattern, importantly the significant CPAC correlations lie in a diagonal orientation

well south of the Equator between 10–20°S and appear more tied to convection within the South Pacific Convergence Zone (SPCZ). The CPAC OLR correlations also lie in a very favorable region of climatological ascent within the SPCZ (Supplementary Fig. 1b), suggesting CPAC convection may be an important mechanism driving high-latitude atmospheric circulation anomalies.

Other significant correlations exist between Larsen C surface melt and tropical variability, including a significant positive SST correlation in the west Pacific warm pool (Fig. 1b) and significant positive OLR correlations in the western tropical Atlantic (Fig. 1d). However, there are no significant correlations with OLR in the west Pacific warm pool, which indicates these SST anomalies that co-vary with Larsen C surface melt are not associated with deep ascent/convection and they are unlikely to be associated with Rossby wave development. In the western tropical Atlantic, the positive OLR correlations lie within a region of climatological subsidence (Supplementary Fig. 1) that suppresses updraft formation, and like the southeast sub-tropical Pacific, this region would not be conducive to Rossby wave development. Moreover, the western tropical Atlantic OLR correlations are broadly associated with negative SST anomalies in the tropical Atlantic, and previous numerical experiments investigating the tropical Atlantic teleconnection to the Southern Hemisphere[34–36] show positive SST anomalies in the tropical Atlantic are associated with warming in the AP. In contrast, the negative tropical Atlantic SST anomalies (and reduced convection) would favor cooling on the AP and therefore are unlikely to be a major contributor to enhanced Larsen C surface melt. We infer from the correlations that CPAC convection is the most likely feature of tropical variability that may be physically connected to Larsen C surface melt due to the strong correlation value of −0.63 and the very favorable environmental conditions conducive for Rossby wave development, and therefore CPAC convection is the focus of the remainder of this study.

The AP climate is known to be influenced by both remote forcing from the tropics and the SAM[37,38]. However, relationships with the tropics are generally confined to the western AP linked to variability in the Amundsen Sea Low and associated thermal advection[31]. Moreover, tropical teleconnections to the Antarctic are generally weak during the austral summer due to weak Rossby wave sources in the sub-tropics and an unfavorable jet stream configuration that inhibits Rossby wave propagation from the tropics into the southern high latitudes[39]. Instead, studies have found the AP climate during summer to be mostly tied to variability in the SAM and the strength and position of the circumpolar westerlies[26,31]. The caveat is that these relationships have largely been derived from a limited number of weather stations confined to the northern tip of the AP[40] (Fig. 1a), over 100 km north of the Larsen C Ice Shelf.

Here we find no significant relationship between Larsen C surface melt and the SAM index during summer (Table 1). Also, the SAM exhibited a weak positive trend during summer (likely from ozone depletion and greenhouse gas increases[41]), which would be associated with warming on the northern Peninsula, while on the contrary, Larsen C surface melt decreased over the same period (Fig. 1g, $p = 0.11$), further indicating SAM and Larsen C surface melt are not linearly related, moreover that positive SAM is not associated with increased Larsen C surface melt. Instead, looking spatially at the atmospheric circulation shows that positive surface melt anomalies are associated with a zonally asymmetric pattern bearing little resemblance to the SAM, and instead is comprised of a wave train of alternating high and low-pressure anomalies emanating from the central tropical Pacific that results in an anomalous anticyclone over Drake Passage (Drake Z500; Fig. 1f, h). Indeed, the decrease in Larsen C surface melt over 1991–2015 is consistent with a weak positive

CPAC OLR trend (reduced convection; Fig. 1e) along with a weak negative Drake Z500 trend (Fig. 1i; $p > 0.10$) over that period.

Larsen C surface melt and CPAC OLR correlations reveal both are associated with a very similar asymmetric circulation pattern over the South Pacific (cf. Fig. 2a, b), and both are strongly associated with a positive pressure anomaly over Drake Passage ($r > 0.60$, $p < 0.01$; Fig. 2c, d). The anticyclonic anomaly over Drake Passage produces moist southwesterly flow across the central and southern AP resulting in significant moisture flux convergence along the southwest AP and moisture flux divergence over the eastern AP producing a classic foehn signature (Fig. 2e, f). The correlations indicate foehn would be particularly favored over the Larsen C ice shelf where the moisture flux divergence anomalies are strongest. These results are consistent with previous studies that found a relationship between Larsen C surface melt and foehn occurrence[20,23,24], and here we show that foehn across the central and southern portions of the eastern AP is tied to moist southwesterly flow associated with an anticyclone in Drake Passage, while SAM variability, known to modulate foehn over the northeast AP, is more associated with westerly to northwesterly flow across the northern tip of the AP[31].

The strong relationship between CPAC convection and Larsen C surface melt is further illustrated when examining the extreme summer/monthly surface melt events (Table 2). For the top three (90th percentile) surface melt summers/months, anomalous deep CPAC convection (CPAC OLR $\leq -0.5\sigma$) occurred in 13 of the 15 events, with one neutral CPAC OLR year and only one positive CPAC OLR year during the February 1998 El Niño (SOI $\leq -0.5\sigma$) and neutral SAM event. There is no consistent SAM phase for extreme melt events: SAM was negative in four cases, positive in four cases, and neutral in the remaining seven cases, consistent with the weak correlation over the full period (cf. Table 1). It is perhaps noteworthy that El Niño conditions (SOI $\leq -0.5\sigma$) were seen for nine of the 15 extreme surface melt events, as this is in contrast to La Niña being associated with anomalously warm temperatures on the northern AP due to a deepening of the Amundsen Sea Low[31,37]. This underscores the crucial role of high pressure over the Drake Passage and the associated southwesterly flow across the AP in producing foehn warming over and south of Larsen C, which is not a feature of La Niña or positive SAM.

**Central tropical Pacific convection connected to record warmth and surface melt.** Next, we investigate the synoptic conditions during two recent extreme foehn events that resulted in record-high surface air temperatures and surface melt on the eastern AP: 24 March 2015, when a record warm temperature for the Antarctic continent (+17.5 °C) was set at Esperanza station (Fig. 3a–c)[21], and the more recent 6 February 2020 event at Esperanza (+18.3 °C) that broke the previous record, and also coincided with record-high surface melt across the AP (Fig. 3d–f)[42–44].

During 24 March 2015 (the pentad centered on 24 March 2015, i.e., mean anomalies for 22–26 March 2015), there was anomalous deep convection in the central tropical Pacific (Fig. 3a; black box; cf Fig. 1d) associated with a record-strong Madden–Julian Oscillation (MJO) event[45]. The anomalous upper-tropospheric streamfunction and horizontal wave activity[46] (Methods) demonstrate anomalous poleward wave fluxes and an associated wave train triggered by the central tropical Pacific convection that produced an upper-tropospheric cyclonic anomaly over the middle-latitude South Pacific (~120°W and 45°S) and a downstream anticyclonic anomaly over Drake Passage that is consistent with the Larsen C surface melt and CPAC OLR interannual correlation patterns. At the surface

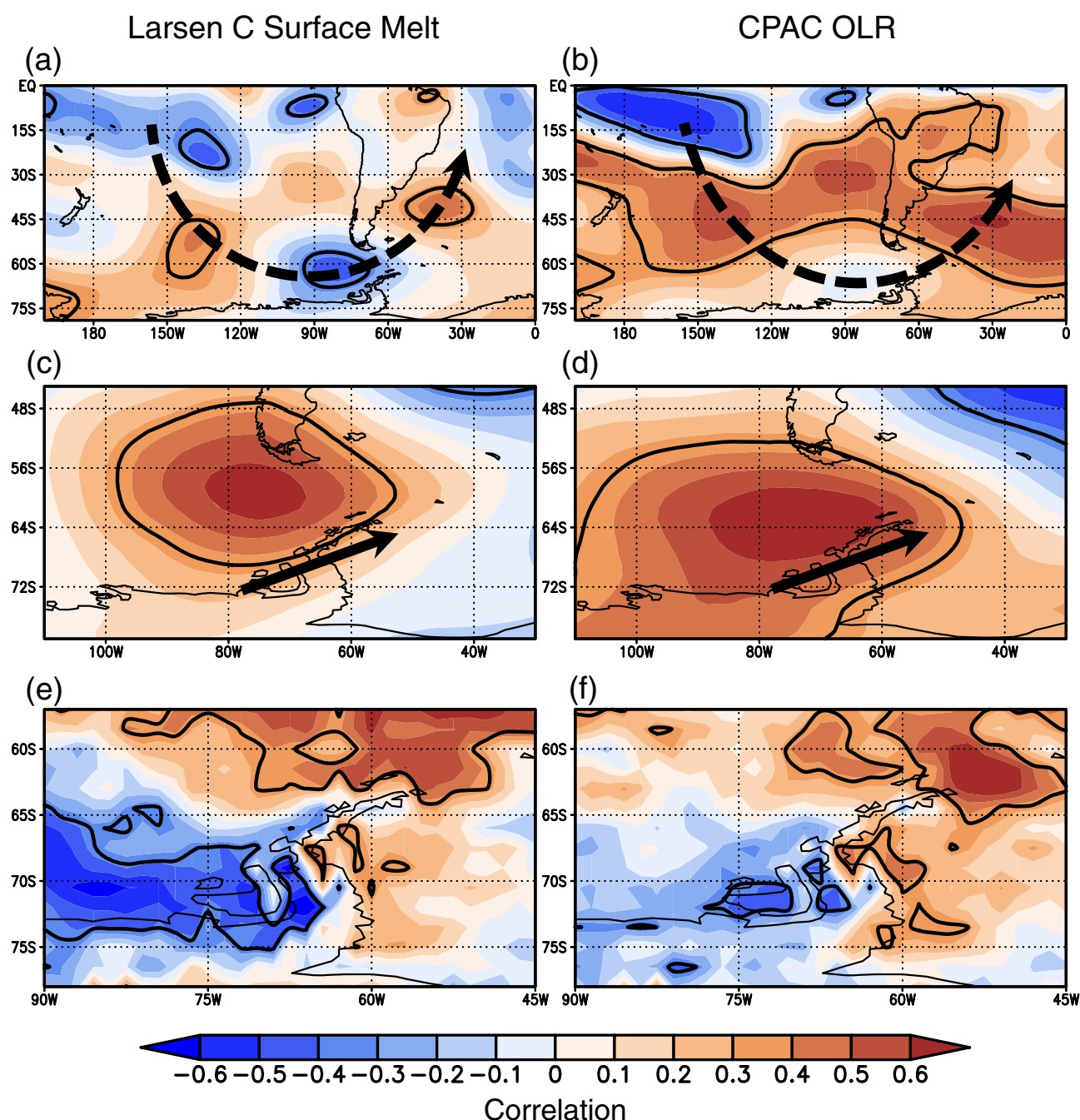

**Fig. 2 The atmospheric circulation anomalies associated with CPAC convection compared to Larsen C surface melt.** The DJFM detrended correlation (shaded), 1991–2015, of (left; (**a**, **c**, **e**) Larsen C surface melt and (right; (**b**, **d**, **f**) CPAC OLR with **a**, **b** 250 hPa streamfunction, **c**, **d** 500 hPa geopotential height, and **e**, **f** vertically integrated moisture flux divergence. Correlations significant at p < 0.10 are denoted by bold black contours. CPAC OLR correlations are multiplied by −1 to reflect anomalies associated with enhanced CPAC convection. Dashed black arrows in **a**, **b** schematically show the path of Rossby wave propagation. Solid black arrows in **c**, **d** schematically show anomalous winds across the AP associated with the Drake Passage anticyclonic circulation anomaly.

(Fig. 3b), this resulted in a large, elongated negative mean sea level pressure (MSLP) anomaly stretching from the middle latitudes poleward into the western Amundsen Sea and a strong positive MSLP anomaly over Drake Passage, which together transported anomalous warm air poleward to the AP and locally produced warm southwesterly flow across the AP. Thermodynamic foehn warming is typically more extreme when warm air masses ascending the windward side are also high in atmospheric moisture, which allows more latent heat to be released and a greater potential temperature to be achieved[47]. Indeed, the

anomalous circulation pattern triggered an "atmospheric river" (AR)[48] (Methods), a corridor of extreme poleward moisture flux, that made landfall on the AP on 24 March 2015 (Fig. 3c) three hours prior to the 24 March 2015 temperature record.

Nearly identical synoptic conditions were seen for the more recent 6 February 2020 record-high temperature and surface melt event[42] (Fig. 3d–f). Again, anomalous deep convection over the central tropical Pacific triggered a wave train across the southeast South Pacific resulting in a cyclonic anomaly in the middle-latitude South Pacific (~120°W and 50°S) and an anticyclone over

**Table 2 The large-scale climate patterns during extreme high Larsen C surface melt.**

| Year | Melt (mm) | CPAC OLR / ENSO / SAM |
|---|---|---|
| **DJFM** | | |
| 1994/95 | 433.7 | −CPAC |
| 1992/93 | 415.2 | −CPAC / EN / −SAM |
| 1997/98 | 411.8 | −CPAC / EN |
| **December** | | |
| 1994 | 229.2 | −CPAC / EN / +SAM |
| 1992 | 224.8 | −CPAC / EN |
| 1996 | 200.7 | LN |
| **January** | | |
| 2006 | 275.4 | −CPAC / LN / −SAM |
| 2007 | 257.0 | −CPAC / EN / −SAM |
| 1999 | 181.6 | −CPAC / LN / +SAM |
| **February** | | |
| 1998 | 139.7 | +CPAC / EN |
| 2006 | 83.4 | −CPAC / −SAM |
| 1993 | 65.7 | −CPAC / EN |
| **March** | | |
| 2015 | 14.5 | −CPAC / EN / +SAM |
| 1993 | 12.2 | −CPAC / EN |
| 1996 | 5.2 | −CPAC / LN / +SAM |

The CPAC OLR, El Niño-Southern Oscillation (ENSO), and SAM conditions for the top three (90th percentile) DJFM and monthly Larsen C surface melt years during 1991–2015. Negative (enhanced convection; CPAC OLR ≤−0.5σ) and positive (suppressed convection; CPAC OLR ≥0.5σ) CPAC OLR years are labeled as −CPAC and +CPAC, respectively. La Niña (SOI ≥0.5σ) and El Niño (SOI ≤−0.5σ) years are labeled as LN and EN, respectively. Positive SAM (SAM ≥0.5σ) and negative SAM (SAM ≤−0.5σ) years are labeled +SAM and −SAM, respectively. Neutral CPAC OLR, SOI, and SAM index years within ±0.5σ are not labeled.

Drake Passage (Fig. 3d). Like the 24 March 2015 event, the elongated "double-barrel" low-pressure anomaly in the South Pacific, with a northern low-pressure center in middle latitudes and a southern low-pressure center in the western Amundsen Sea (Fig. 3e), coupled with the Drake Passage anticyclone, resulted in strong poleward heat transport from low-latitudes that brought positive temperature anomalies of more than 6 °C across nearly all of West Antarctica and the AP as well as a southwesterly flow across the AP along the southern edge of the Drake Passage anticyclone. With the lower latitude cyclonic anomaly sourcing a warm moist air mass from the sub-tropics, an AR was again triggered that made landfall on the western AP nine hours prior to the 6 February 2020 record-high temperature (Fig. 3f). In addition to setting the new record temperature, the AP experienced its highest surface melt on record for early February with surface melt affecting more than 50% of the region[42].

**Attributing the atmospheric response to central tropical Pacific convection.** The correlation and case study results presented thus far suggest CPAC convection is likely a key driver of the forced asymmetric circulation pattern responsible for record warm events and interannual variability in Larsen C surface melt. However, it is possible the circulation pattern could be caused by other regions in the tropics. Therefore, we next isolate the direct effect of CPAC convection on the atmosphere by performing a sensitivity experiment with a global climate model (Methods). We compare two 30-year simulated December-February (DJF) climatologies forced with annually repeating global climatological SSTs with and without a surface heating anomaly in CPAC (Methods); i.e., all other regions in the tropics are set to their climatological state and the only difference between the two simulations is a CPAC perturbation (Fig. 4).

The CPAC surface heating anomaly generates a local increase in deep convection in CPAC (color shading in Fig. 4a), consistent with the favorable climatological ascent across this region. The

convection forces an anomalous atmospheric circulation pattern across the South Pacific that mirrors both the Larsen C surface melt/CPAC OLR correlation patterns and the synoptic conditions during the two record-high temperature events (cf. Figs. 1d, 3a, d). In particular, the CPAC perturbation produces a deep, elongated cyclone across the South Pacific centered near ~120°W, 50°S (crucial for sourcing a warm and moist low-latitude air mass) stretching poleward into the western Amundsen Sea coupled with an anomalous anticyclone over Drake Passage (crucial for steering the warm moist air southwesterly across the AP). The anomalous wave flux shows CPAC convection forces two regions of poleward wave propagation: the strongest being immediately downstream/southeast of the CPAC convection which would aid in producing the northern mid-latitude low-pressure center, and a second weaker region near New Zealand. There is clear wave refraction off the summertime mid-latitude jet (gray shading in Fig. 4a) in the region southeast of CPAC, while the wave activity propagates deeper into high-latitudes southeast of New Zealand reaching the Amundsen Sea. Together, the two regions of wave propagation merge over the central South Pacific to produce the large elongated cyclonic anomaly, and with continued eastward wave propagation into Drake Passage that builds high pressure there. Locally over the AP, there are significant increases in precipitation (Fig. 4c) and near-surface warming (Fig. 4d) along the southwest AP and adjacent West Antarctica, reproducing the classic foehn signature associated with surface melt on Larsen C.

The anomalous upper-tropospheric divergent winds (Fig. 4b) together with the climatological jet stream (gray shading in Fig. 4a) provide clues to the development of two distinct regions of wave propagation into high latitudes, which is unusual for summer[39]. Rossby wave sources occur in regions where the divergent wind advects absolute vorticity and where vorticity is generated by vortex stretching from upper-level divergence. By definition, Rossby wave sources maximize where strong divergent flow intersects jet streams and near exit/entrance regions of jet cores where there is strong localized divergence. The summertime mid-latitude jet simulated by CESM is comprised of two distinct jet streaks (dark gray shading in Fig. 4a): one in the Atlantic and Indian (Indo-Atlantic) Ocean, and a second in the South Pacific along the northern edge of the Amundsen Sea Low. With divergence being maximized in the entrance and exit regions of jet streaks, the strongest poleward wave propagation into high latitudes is seen near the exit region of the Indo-Atlantic jet streak and near the entrance region of the South Pacific jet streak (black circles in Fig. 4b), despite the latter being dominated by wave refraction further to the west. Therefore, we hypothesize that CPAC-triggered planetary wave activity propagates deep into high latitudes via these two anomalous Rossby wave sources, one in which the divergent flow intersects the exit region of the Indo-Atlantic jet streak south of New Zealand, with eastward wave propagation along the mid-latitude jet that deepens the Amundsen Sea Low at high latitudes, and a second immediately downstream/southeast of CPAC where strong poleward wave fluxes produce the primary mid-latitude cyclonic anomaly, and a localized Rossby wave source in the entrance region of the South Pacific jet streak allows continued wave propagation into Drake Passage that builds the anticyclone. These simulation results are also consistent with previous findings highlighting the role of Rossby wave teleconnections in shaping extreme warming and surface melt events on the AP[30,45].

**Drivers of central tropical Pacific convection and its connection to atmospheric rivers.** In this final section, we investigate what triggers CPAC convection during the austral summer

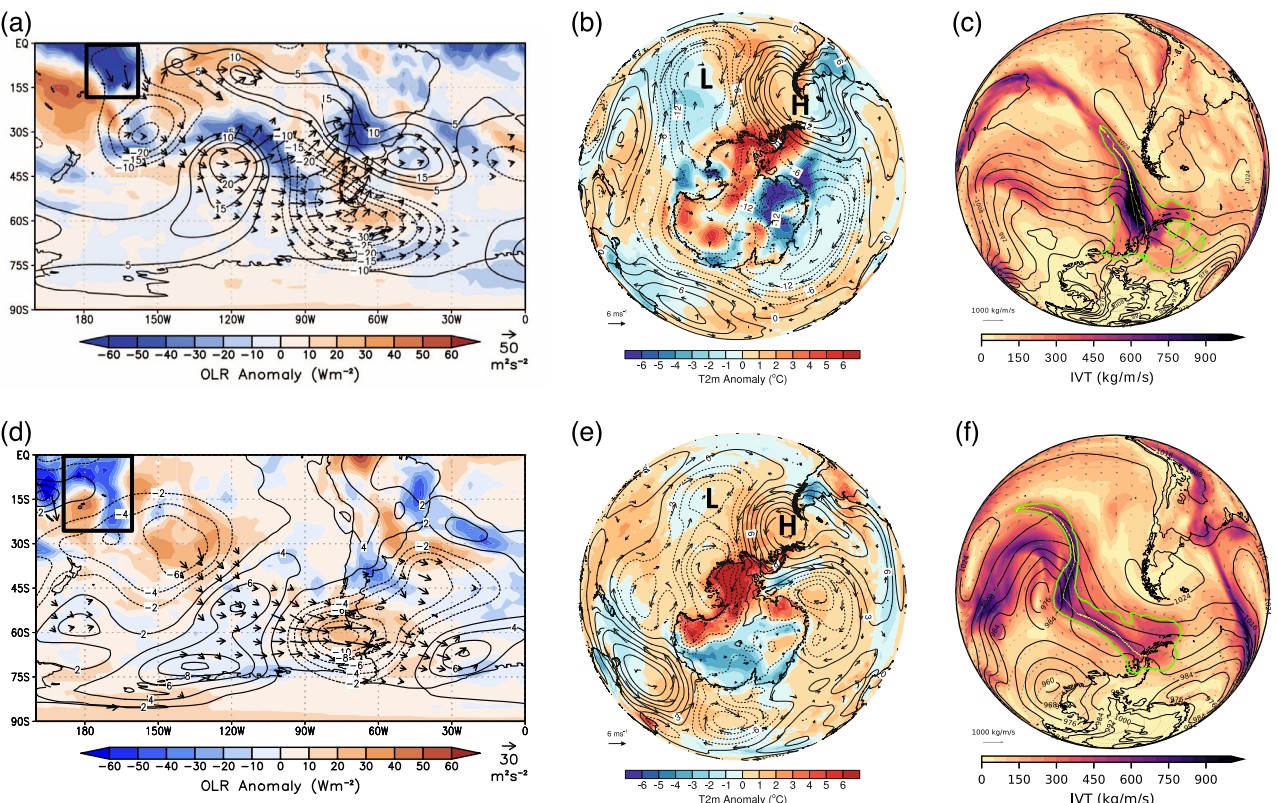

**Fig. 3 The CPAC-forced circulation anomalies driving extreme high temperatures on the eastern Antarctic Peninsula.** (top) The 24 March 2015 and (bottom) 6 February 2020 record-high temperature events. **a** The OLR (shaded), 200 hPa streamfunction (contoured) and 200 hPa stationary wave flux (vectors) anomalies and **b** surface air temperature (T2m, shaded), MSLP (contoured), and 10 m wind (vectors) anomalies for 22–26 March 2015, and **c** the vertically integrated moisture flux (IVT; shaded, vectors), MSLP (contours), and outline of the landfalling AR (green contour; IVT exceeding 85th percentile) and AR axis (yellow line; a pathway of maximum IVT) (Methods) at 06 UTC 24 March 2015. **d**, **e** are as in **a**, **b** except for 4–8 February 2020 and **d** is for the 850 hPa level, and (**f**) is as in (**c**) except for 06 UTC 6 February 2020. The CPAC OLR region is denoted with a black box in the upper left corner of (**a**, **d**), and the location of the high and low-pressure centers is given as an "H" and "L", respectively, in (**b**, **e**).

(Fig. 5). On interannual time scales, CPAC convection is broadly associated with anomalous deep convection across the central equatorial Pacific reminiscent of El Niño and/or the MJO, but importantly, it is most strongly associated with an off-equatorial diagonal band of convection more characteristic of the SPCZ (Fig. 5a, b). CPAC convection is associated with a sub-tropical surface cyclone located southeast of the CPAC region (Fig. 5c, d) and strong southerly cold air advection into CPAC on the western side of the cyclone (Fig. 5e, f), with the diagonal band of convection being located along the surface cold front/baroclinic zone. Similarly, several days preceding the two recent record AP temperature events (Supplementary Fig. 2), a sub-tropical cyclone developed south of CPAC and an associated cold front advanced northward into the central tropical Pacific triggering a band of intense convection. CPAC convection is also broadly associated with positive SST anomalies across the central tropical Pacific reminiscent of El Niño (Fig. 5g, h), however, the strongest SST correlations are located along the northern edge of CPAC and across equatorial latitudes rather than directly beneath CPAC.

We infer that SST anomalies (i.e., surface heating) are not the primary trigger of the convection, but rather cold frontal intrusions from sub-tropical and middle latitudes are the dominant mechanism, particularly in triggering the off-equatorial, diagonal nature of CPAC convection. Indeed, convection in the CPAC region (i.e., in the eastern SPCZ) is well-known to be highly transient and influenced by mid-latitude wave activity[49]. However, background positive SST anomalies, as well as a convectively "active" MJO phase, would produce favorable

environmental conditions conducive for intense convective development along the northward advancing cold front, and we infer that these mechanisms likely play a conditional, background role. This is supported by the correlation (Fig. 1b) and composite (Table 1) analyses, both of which show no significant preference for an SST anomaly state across the tropical Pacific during anomalous CPAC convection. And while the MJO was strongly active during the March 2015 event[45], it was only weakly active during the February 2020 event (not shown).

Lastly, ARs have been shown to be a significant driver of extreme surface melt events across the West Antarctic region[47], as seen in the two record AP temperature and surface melt events (Fig. 3). Here we investigate if AR activity is associated with Larsen C surface melt on interannual time scales and if CPAC convection is an important driver of this interannual variability (Fig. 6). Consistent with previous findings[47], we find a strong statistically significant ($p < 0.01$) correlation of 0.79 between total summer Larsen C surface melt and the total number of extreme AP landfalling ARs (Methods) during summer (Fig. 6a). We also find strong, significant ($p < 0.01$) correlations between the total number of extreme landfalling ARs and Drake Z500 and CPAC OLR (0.64 and −0.70, respectively; Fig. 6b, c). Therefore, the frequency of extreme summertime landfalling ARs plays an important role in the total amount of summer surface melt on Larsen C, and CPAC convection and its associated Drake Passage anticyclone are key features governing the number of extreme landfalling ARs. A similar connection is seen on synoptic (daily) time scales (Fig. 6d), with a significant increase in AR frequency

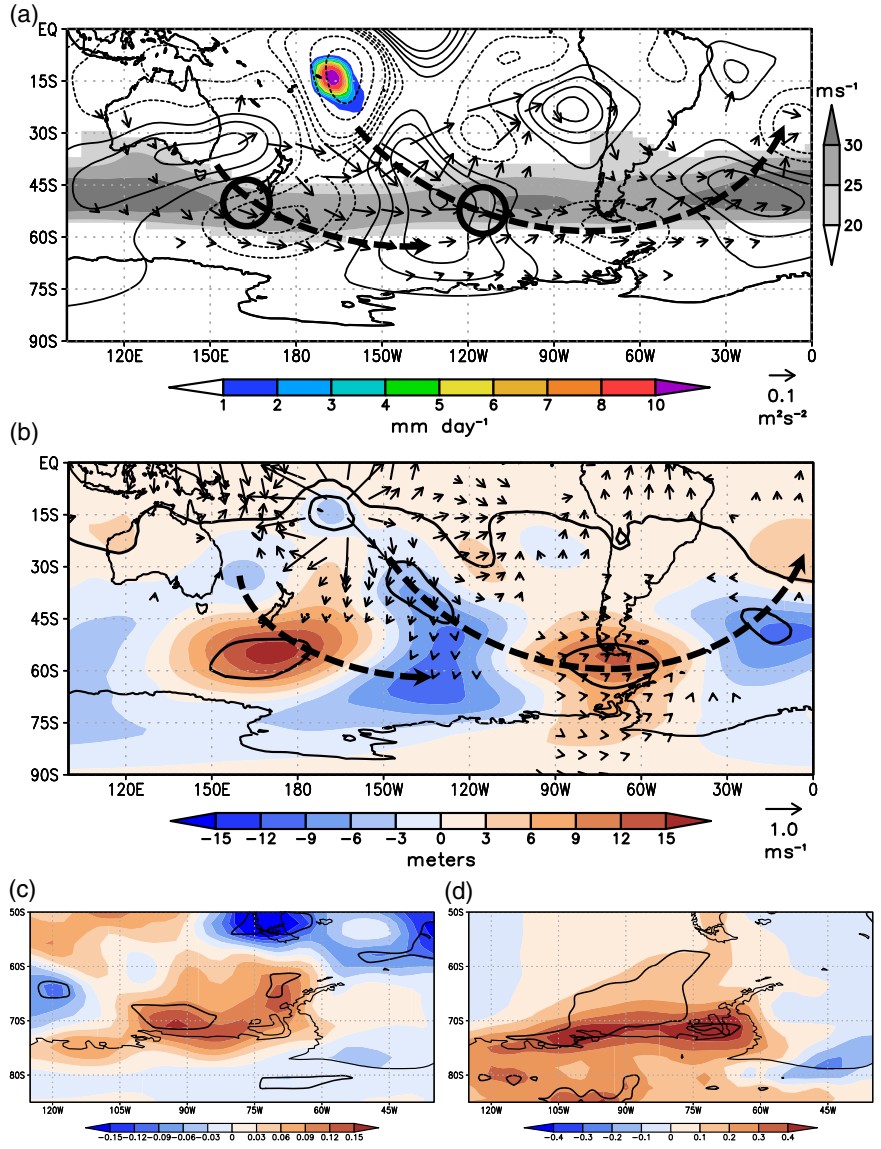

**Fig. 4 The simulated atmospheric response to CPAC convection.** The difference in CPAC perturbation minus control simulated 30-year climatologies for DJF. The difference in **a** total precipitation (color shaded), 200 hPa streamfunction (contour), and 200 hPa wave flux (vectors); **b** 500 hPa geopotential height (shaded) and 200 hPa divergent wind (vectors); **c** total precipitation, and **d** surface air temperature. In **a**, the 30-year climatological 300 hPa zonal wind from the control run is shaded in gray and the black circles schematically show the two regions of divergence in the exit region of the Indo-Atlantic jet streak and entrance region of the Pacific jet streak. In **a**, **b**, schematic arrows are drawn in black showing the two wave train paths. In **b**–**d**, bold black contours denote differences that are statistically significant at $p < 0.10$, and vectors in (**b**) are drawn only if at least one component of the divergent wind is significant at $p < 0.10$.

across the Bellingshausen Sea and southwest AP during days with strong CPAC convection (CPAC OLR $\leq -0.5\sigma$), with a maximum increase in AR days (15–20% increase compared to the DJFM AR climatology) over the central AP near Larsen C.

These findings demonstrate that CPAC convection is a key driver of both interannual variability in Larsen C surface melt during the summer melt season and the occurrence of extreme high surface temperatures and surface melt events than can trigger ice shelf collapse. The zonally asymmetric circulation pattern triggered by CPAC convection is distinctly different from that of SAM, suggesting the projected positive SAM trend over the remainder of this century due to increasing greenhouse gases may be less important in governing the fate and potential collapse of Larsen C (and the remaining section of Larsen B) than future variability in CPAC convection. We find CPAC convection is

primarily caused by baroclinic mid-latitude wave activity and associated cold frontal intrusions, and therefore future variability in ENSO or other tropical climate modes, such as the Interdecadal Pacific Oscillation, may not have a significant influence on future CPAC convective events. An improved understanding of future CPAC variability, and more generally the occurrence of asymmetric circulation patterns comprising elongated cyclones stretching into low-latitudes situated alongside strong anticyclones (and associated AR activity), will help support more reliable projections of future Antarctic high-temperature extremes, surface melt, and ice shelf stability.

## Methods

**Data**. Atmospheric circulation was investigated with the European Centre for Medium-Range Weather Forecasts (ECMWF) fifth generation atmospheric

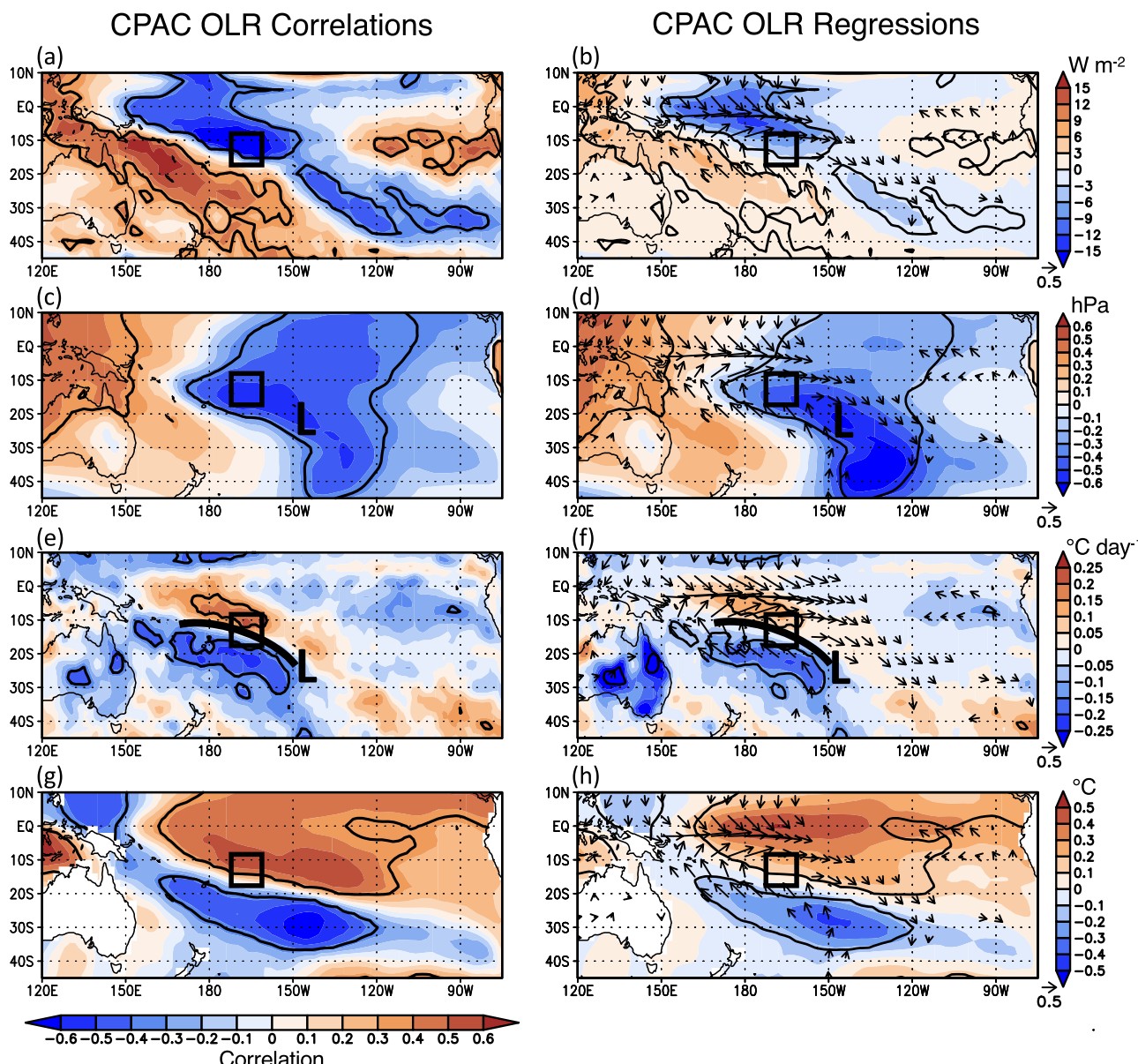

**Fig. 5 The local synoptic conditions governing CPAC convection.** The detrended DJFM (left) CPAC OLR correlations and (right) standardized CPAC OLR regressions with **a**, **b** OLR, **c**, **d** MSLP, **e**, **f** 925 hPa temperature advection, and **g**, **h** SST during 1991–2015. Also shown on the right are regressions with 10 m wind for (**b**, **d**, **h**) and 925 hPa wind for (**f**). The correlations/regressions are multiplied by −1 to show conditions associated with enhanced CPAC convection. The bold contours denote correlations and regressions significant at $p < 0.10$, and wind vectors are shown only if at least one regression component is significant at $p < 0.10$. The CPAC area (10–15°S, 170–165°W) is denoted by the black box, and the position of the surface low-pressure center and its cold front are denoted by an "L" and curved black line, respectively.

reanalysis ERA5[50]. ERA5 replaces the ECMWF Interim (ERA-Interim) reanalysis, which was considered to be the best reanalysis for depicting Southern Hemisphere high-latitude climate[51], with improved model physics, core dynamics, and data assimilation as well as a higher horizontal resolution. We use monthly-mean ERA5 fields to compute the seasonal means employed in the correlation analysis, and 6-hourly fields to compute the 5-day group means investigated for the two case studies. Anomalies for the two pentads in Fig. 3a, b, d, e are from the 1981–2010 climatological mean.

Tropical variability is investigated using monthly-mean sea surface temperature (SST) data from the National Oceanic and Atmospheric Administration (NOAA) Extended Reconstructed SST Version 5 (ERSSTv5) dataset[52], which has a horizontal grid spacing of 2° × 2°. We investigate variability in tropical deep convection using the monthly and daily-mean fields of outgoing longwave radiation (OLR) from the NOAA Interpolated OLR dataset[53], which has a 2.5° × 2.5° horizontal grid spacing. Both the ERSSTv5 and OLR datasets were obtained online from the NOAA Physical Sciences Laboratory (https://psl.noaa.gov/data/gridded/).

Variability in the El Niño-Southern Oscillation is investigated using the Southern Oscillation Index (SOI), which is the difference in standardized mean sea

level pressure anomalies between Tahiti (eastern tropical Pacific) and Darwin, Australia (west Pacific warm pool), obtained from the NOAA Climate Prediction Center (https://www.cpc.ncep.noaa.gov/data/indices/). The Southern Annular Mode (SAM) is investigated using the observation-based index of[25], which is the difference in standardized zonal-mean sea level pressure between 40°S and 65°S. Variability in central tropical Pacific (CPAC) deep convection is investigated using the OLR area-averaged over the region 10–15°S, 170–165°W. Circulation variability over Drake Passage (Drake Z500) is investigated using the area-averaged ERA5 500 hPa geopotential height over the region 57–62°S, 81–71°W. All area averages are weighted by the cosine of the latitude.

**Larsen C ice shelf surface melt**. Surface melt on the Larsen C Ice Shelf was derived from a modeling experiment using the Polar Weather Research and Forecasting Model (Polar-WRF version 3.9.1) over the period December 1991 to March 2015. Polar-WRF is a state-of-the-art limited-area mesoscale modeling system[54], which is a polar-optimized version of the WRF model[55]. The model includes modified land-surface model sea ice representation as well as enhanced

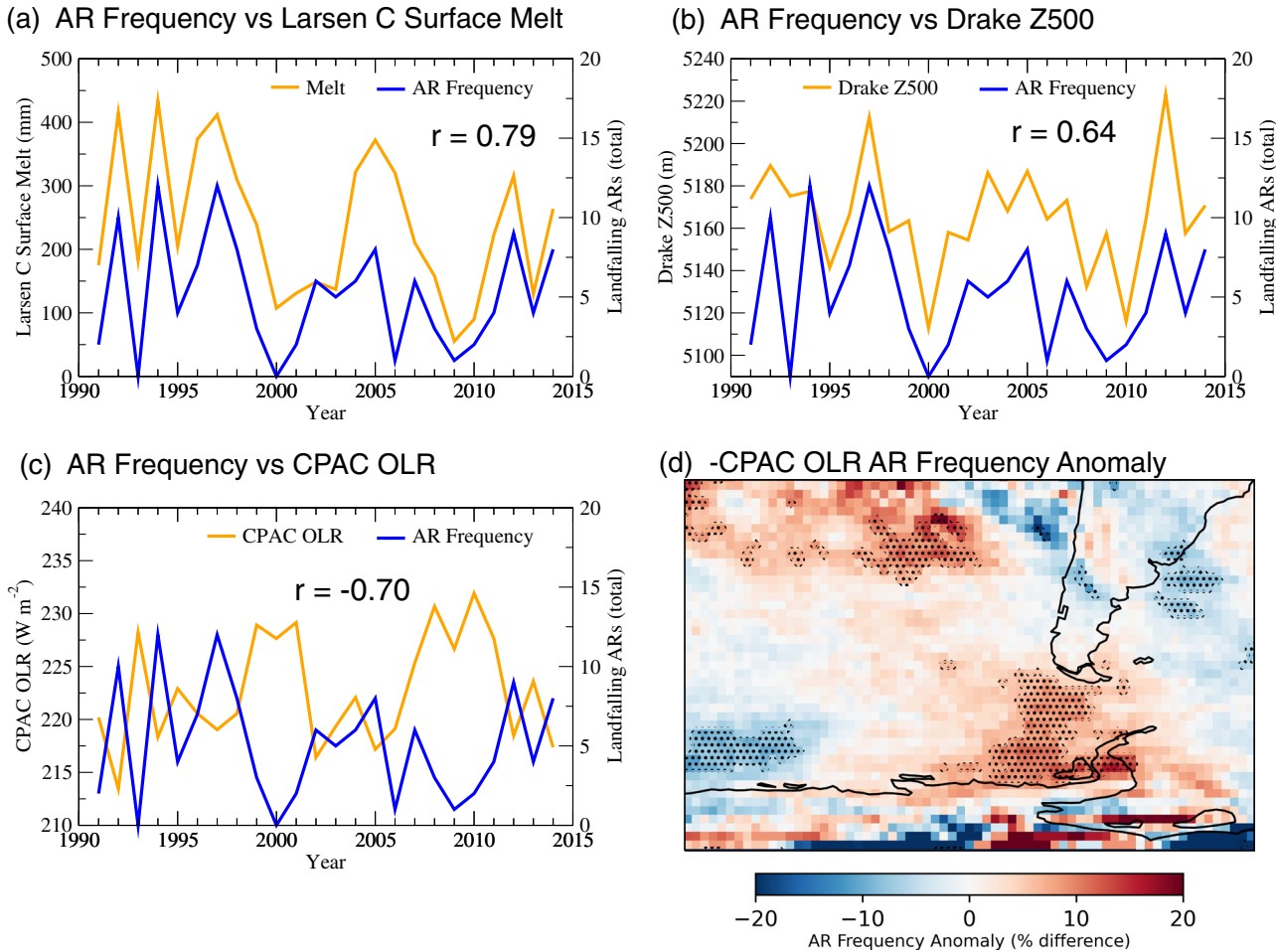

**Fig. 6 CPAC convection a key driver of extreme atmospheric rivers on the Antarctic Peninsula. a–c** Timeseries of (blue) the total number of DJFM extreme landfalling ARs alongside the (orange) DJFM **a** Larsen C total surface melt, **b** Drake Z500, and **c** CPAC OLR. Inset is the correlation coefficient between the two timeseries. **d** Composite anomaly of AR frequency for anomalous CPAC convection days (daily CPAC OLR ≤ −0.5σ) showing the percentage difference relative to the DJFM AR frequency climatology. Stippling in (**d**) denotes anomalies significant at $p < 0.10$.

treatment of the snowpack, sea ice, and cloud radiative processes over the polar regions. Polar-WRF is the base model of the Antarctic Mesoscale Prediction System (AMPS) that is operationally run by the National Center for Atmospheric Research (NCAR), USA[56]. More details of the Polar-WRF and its components can be found in refs. [54,57,58] and citations therein.

Our modeling experiment was conducted over the Antarctic Peninsula with two nested domains at 0.4° (~45 km) and 0.13° (~15 km) spatial resolutions on a polar stereographic projection. Initial and lateral boundary conditions including SST fields and initial soil parameters are provided by ERA-Interim at 6-hourly intervals with a grid spacing of 0.75° × 0.75°. The sea-ice data are based on the 25-km resolution Bootstrap dataset[59]. Land-type and topography information for the model are from the United States Geological Survey land-use data and GTOPO30 elevation data, respectively. In this study, we use only 15-km simulation outputs from the inner domain, which is centralized on the Antarctic Peninsula and Larsen Ice Shelf and has 208 × 190 grid cells. This domain employs 61 vertical levels between the surface and the model top at 10 hPa.

The simulations were performed using: (1) the new version of the rapid radiative transfer model[60] for general circulation models (RRTMG) for both shortwave and longwave radiations, (2) the Morrison double-moment microphysics scheme[61], (3) the Mellor–Yamada–Janjic (MYJ) boundary layer scheme[62], (4) the Grell−Freitas ensemble cumulus scheme[63], and (5) the Noah-MP land-surface model[64,65]. More information about the model configuration and setup as well as its ability to capture recent temperature trends can be found in ref. [66].

Surface melt is a direct output from the model and is provided as accumulated liquid equivalent melted snow. Therefore, surface melt rate was calculated for each daily time step (i.e., (melt($t = t + 1$) - melt($t$))/dt) using the accumulated melted snow variable. Daily surface melt values (mm day$^{-1}$) were then summed to get the total monthly and seasonal surface melt. A mask polygon shapefile of Larsen C Ice Shelf was created for masking. Therefore, only grid points within the Larsen C Ice Shelf shapefile were used for calculating total monthly and seasonal surface melt.

**Stationary wave flux activity**. The horizontal wave activity flux, **W**, formulation of[46] was employed to examine anomalous stationary wave activity. The zonal (**Wx**) and meridional (**Wy**) components take the form:

$$Wx = \frac{p\cos(\phi)}{2|\mathbf{U}|}\left[\frac{\bar{U}}{a^2\cos^2(\phi)}\left(\left(\frac{\partial\psi}{\partial\lambda}\right)^2 - \psi\frac{\partial^2\psi}{\partial\lambda^2}\right) + \frac{\bar{V}}{a^2\cos(\phi)}\left(\frac{\partial\psi}{\partial\lambda}\frac{\partial\psi}{\partial\phi} - \psi\frac{\partial^2\psi}{\partial\lambda\partial\phi}\right)\right]$$

$$Wy = \frac{p\cos(\phi)}{2|\mathbf{U}|}\left[\frac{\bar{U}}{a^2\cos(\phi)}\left(\frac{\partial\psi}{\partial\lambda}\frac{\partial\psi}{\partial\phi} - \psi\frac{\partial^2\psi}{\partial\lambda\partial\phi}\right) + \frac{\bar{V}}{a^2}\left(\left(\frac{\partial\psi}{\partial\phi}\right)^2 - \psi\frac{\partial^2\psi}{\partial\phi^2}\right)\right]$$

where $\lambda$ and $\phi$ are the longitude and latitude coordinates, respectively, $\psi$ is the geostrophic streamfunction anomaly, $\bar{U}$ and $\bar{V}$ are the mean climatological zonal and meridional winds, respectively, $|\mathbf{U}|$ is the magnitude of the climatological horizontal winds, $p$ is the normalized pressure, which is the pressure divided by a standard reference pressure of 1000 hPa, and $a$ is the radius of the Earth. Defined in this fashion, the fluxes indicate the direction of anomalous horizontal stationary Rossby wave propagation[46,67].

**Detection of atmospheric rivers**. In Fig. 6 we investigate co-variability between atmospheric river (AR) activity, CPAC convection, and Larsen C surface melt using the AR catalog V3.0 of[68] based on ERA-Interim, obtained from https://ucla.app.box.com/v/ARcatalog. AR shape boundaries are calculated 6-hourly between 1979–2019, at 1.5° × 1.5° spatial resolution; we convert to daily AR frequency if an AR was detected during any of the 6-hourly time-steps during that day. In Fig. 6a–c, we define the total number of extreme landfalling ARs during DJFM (1991–2014/15) as the number of days when an AR intersects the AP grid area of 61.5–72°S, 75–63°W. "Extreme" landfalling AR events are defined as AR landfall days with daily-mean integrated water vapor transport (IVT) area-averaged over the AP exceeding the 95$^{th}$ percentile (based on the DJFM IVT climatology from ERA-Interim[69]). To calculate AR frequency in Fig. 6d, for each day and grid cell we

record a value of 100 if an AR is detected, otherwise we return a value of zero. Thus, the mean of AR frequency over any time period can be interpreted as the percentage of days where AR conditions are observed (i.e., AR frequency has units of "% of days"). Figure 6d shows composite anomalies of AR frequency for days of anomalous CPAC convection (daily-mean CPAC OLR ≤−0.5σ from the DJFM CPAC OLR climatology over 1979–2019). The anomalies show the percentage difference in AR frequency relative to the mean DJFM AR frequency from 1979 to 2019.

Because the AR catalog of[68] ends in 2019, we employed our own AR detection algorithm to investigate the 24 March 2015 and 6 February 2020 record temperature events using a modified version of the[70] method. Identification of ARs is based on the intensity and geometry of IVT sourced from the ERA5 reanalysis dataset. The algorithm uses 6-hourly IVT data with a horizontal grid resolution of $0.25° \times 0.25°$. The algorithm identifies regions of enhanced IVT whose shape and IVT direction are consistent with the AR definition. Regions of enhanced IVT are defined where IVT magnitude in a given month exceeds the 85th percentile IVT for the 3-month period centered on that month. A fixed lower limit of 150 kg m$^{-1}$ s$^{-1}$ is also imposed. Contiguous regions of grid cells with IVT values above the percentile threshold and fixed lower limit are isolated and labeled. An AR is considered to be a landfalling AR if a region of enhanced IVT intersects a grid cell occupied by the Antarctic Peninsula. The object axis is then calculated following[71]. Axis calculation can be described as follows:

1. The landfall location is labeled as the target grid cell, $e$;
2. The IVT direction at $e$ is calculated and discretized into one of eight cardinal directions (N, NE, E, SE, S, SW, W, NW). Of the eight grid cells adjacent to $e$ the upstream grid cell, $s$, and the two grid cells neighboring $s$ are identified. Of these three candidate grid cells, the one with maximum IVT is tested to determine if the IVT exceeds the percentile threshold;
3. If the IVT threshold is exceeded this grid cell is labeled as the new target grid cell $e$, and we repeat step 2. This process is continued until the upstream grid cell fails to exceed the threshold or a grid cell is detected twice.

The length of the object is computed as the sum of the distances between neighboring axis cells. To be identified as an AR, the following geometry criteria must be satisfied:

1. Length check: The length of the object must exceed 2000 km.
2. Narrowness check: The width of an object is defined as its surface area divided by its length. An object is discarded if its length/width ratio is less than 2;
3. Mean meridional IVT criterion: An object is discarded if the mean IVT does not have a poleward component greater than 50 kg m$^{-1}$ s$^{-1}$. This filters objects that do not transport moisture toward higher latitudes.
4. Coherence in IVT direction criterion: An object is discarded if more than half of the grid cells have IVT deviating more than 45 degrees from the object's mean IVT. This filters objects that do not feature a coherent IVT direction.

**Central tropical Pacific sensitivity experiment**. To investigate the atmospheric response to anomalous convection in the central tropical Pacific, we performed a sensitivity experiment using the NCAR Community Earth System Model version 1.2[72]. The model was run in atmosphere-only mode using Community Atmosphere Model 5 (CAM5) physics and dynamics with a horizontal resolution of $1.9° \times 2.5°$ and 30 vertical levels. Pre-industrial concentrations of greenhouse gases and stratospheric ozone representative of the 1850s are prescribed along with climatological monthly-mean sea ice concentrations (1982–2011) and SSTs (1950–2017). Following a one-year spin up, two 30-year simulations were performed: a control simulation with annually repeating global climatological SSTs and a CPAC perturbed simulation in which a $+2$ °C SST anomaly is applied to the central tropical Pacific centered at 168°W, 13°S, and is dampened to zero following a sine function moving away from the anomaly center in a $6° \times 6°$ box. The SST heating anomaly generates a local increase in deep convection/rainfall in the region of CPAC OLR that is significantly correlated with DJFM Larsen C surface melt, and where anomalous deep convection was observed during the 24 March 2015 and 6 February 2020 case studies. We examine the difference in 30-year climatological atmospheric circulation between the perturbed and the control simulation, thus revealing the direct effect of central tropical Pacific convection on the atmosphere. We focus on the December-February (DJF) season only as this is when surface melt on Larsen C is strongest and most frequent, and to narrow our investigation of Rossby wave propagation during the months when previous studies suggest wave propagation from the tropics into the southern high latitudes is strongly inhibited[39].

**Statistical methods**. Correlations and linear trends for the period 1991–2015 were computed using seasonal anomalies based on a standard least-squares method. All data were detrended prior to calculating the correlations. Statistical significance of the correlations and trends was calculated using a two-tailed Student's $t$-test with 22 ($n$-2) degrees of freedom and a null hypothesis that the correlation/trend is zero[73]. The confidence $p$ value was computed by comparing the critical values of the Student's $t$ distribution against the $t$ ratio, which is the least-squares linear

regression compared to the standard error. The significance of the anomalies in the climate model experiment was computed using a two-tailed difference in means (the perturbed minus the control 30-year climatologies) $t$-test with 58 ($n + m - 2$) degrees of freedom and a null hypothesis that the difference is zero. The confidence $p$ value is computed by comparing the standard error of the groups with the critical values of the Student's $t$ distribution. The significance of the AR frequency anomalies is computed using a two-tailed Student's $t$-test for the null hypothesis that the expected value is equal to the mean.

## Data availability
ERA5 data were available online from the Copernicus Climate Change Service (C3S) Climate Date Store (https://cds.climate.copernicus.eu/#!/search?text=ERA5&type=dataset). ERSSTv5 and OLR data were available online from the NOAA Physical Sciences Laboratory (https://psl.noaa.gov/data/gridded/). Output from the Polar-WRF and CESM experiments are available from the authors upon request. The AR catalog V3.0 of[68] is available online from https://ucla.app.box.com/v/ARcatalog. The AR data sourced from the ERA5 reanalysis dataset are available from the authors upon request.

## Code availability
The correlation and difference in means significance values were computed in Fortran, the **W** flux was calculated using the Grid Analysis and Display System (GrADS), and most of the plots were generated using GrADS or Python. Surface melt was masked and processed using the Quantum Geographic Information System (QGIS) and NCAR Command Language (NCL). The AR detection algorithm for the case studies was developed in Python. All code used to perform the calculations and generate the plots are available from the corresponding author upon request.

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

## Acknowledgements

K.R.C. acknowledges funding from the Royal Society of New Zealand Marsden Fund grant MFP-VUW2010 and New Zealand Antarctic Science Platform Development Fund grant ANTA1801. We thank B. Lintner and the Rutgers Office of Advanced Research Computing for access to computing facilities to carry out the CESM simulations. D.B.

acknowledges support from ANID-CONICYT-PAI-77190080, ANID-PIA-Anillo INACH ACT192057, ANID-FONDECYT-11200101, and COPAS COASTAL ANID FB210021. Powered@NLHPC: This research was supported by the supercomputing infrastructure of the NLHPC (ECM-02). The authors thank Amazon Web Services (AWS) for the grants PS_R_FY2019_Q1_CR2 and PS_R_FY2019_Q2_CR2 to execute the Polar-WRF simulations on the AWS cloud infrastructure. We thank J. C. Maureira for carrying out the Polar-WRF simulations and F. Muñoz, N. Valdebenito, and M. Del Hoyo at Center for Climate and Resilience Research (CR)2 for post-processing the Polar-WRF simulations.

## Author contributions

K.R.C. and D.B. conceived and directed this work with contributions from D.K., J.C.K., and J.T. K.R.C. carried out the correlation, regression, and wave flux analyses, performed the CESM experiments, and led the writing of the manuscript. D.B. computed the Larsen C surface melt, carried out the extreme landfalling AR analysis in Fig. 6a–c, and contributed to discussing the results and editing the manuscript. D.K. carried out the AR analysis in Figs. 3c, f, 6d and contributed to discussing the results. J.C.K. and J.T. contributed to discussing the results and editing the manuscript.

## Competing interests

The authors declare no competing interests.
