## [Peer Review File · Nature Communications]

Central tropical Pacific convection drives extreme high temperatures and surface melt on the Larsen C Ice Shelf, Antarctic PeninsulaEditorial Note: Parts of this Peer Review File have been redacted as indicated to remove third-party material where no permission to publish could be obtained.

REVIEWER COMMENTS

Reviewer #1 (Remarks to the Author):

This paper examines the extratropical response in the SH to tropical SST changes with an emphasis on extreme high temperatures and surface melt on the Larsen ice shelf, focusing on the influences of CP convection on atmospheric circulation around the AP. A series of sensitivity experiments are performed to examine this idea, and diagnostic analyses are used to show circulation and wave activity changes proposed to give rise to the response. The experiments and analyses are interesting, and the suggested dynamical mechanisms seem reasonable. The main comments below are aimed at helping to make the messages of the manuscript more concrete and accurate. It is very interesting that the two extreme AR events are connected with strong MJO bursts in the deep tropics on shorter time scales, which is consistent to the connections on interannual time scales. I think this study could be more useful and have more impact if the authors could do more to understand how AR activity is linked to the wave train pattern on interannual time scales. I believe the paper merits publication in Nature communications after minor revision.

1. Impacts of anthropogenic forcing and the stratospheric ozone depletion over the region are not discussed in the paper. I like to hear more thoughts from the authors about how to understand the role of external emission forcing (CO₂ and the ozone depletion) in determining this tropically-driven framework over the period. How do CMIP5 or 6 historical simulations replicate observed surface warming and ice shelf instability over the AP over the past decades (any studies have looked at this before)? Is this tropical-extratropical connection internally generated or externally driven? Without some additional thoughts on this aspect, I am not sure how to use what we learn from this study to guide us improve the future climate projection over the region?

2. Is the tropical mode identified here similar to the CP EL Nino or other SST modes, such as the IPO or PDO? More analysis could be done to shed more light on the origin of this tropical convection over the CP.

3. Seasonal mean AR occurrence over the AP can be calculated based on the ERA5 AR dataset. If so, it is worthwhile to further calculate the connection between this AR occurrence and the circulation pattern (the high-pressure center around the AP) or the tropical mode on year to year time scales, as the authors have done in Fig. 2. The idea of the test is to examine how the overall activity of AR over the region throughout the season responds to large scale circulation forcing.

Reviewer #2 (Remarks to the Author):

This study attempts to build a linkage between the atmospheric deep convection over the central Pacific and the recently observed high-temperature over the Larsen ice shelves in austral summer. By using statistical analysis and numerical model simulations, the authors reveal that deep convection over the central Pacific may generate a stationary Rossby wave train, which further forms an anomalous high-pressure center west to the Peninsula, driving westerly wind anomalies, transporting heat and moisture from lower latitudes, and producing strong surface warming and melting.

Over all I think that this is a very interesting study, and may potentially be an important contribution to the field. While the accelerated ice shelf melting around West Antarctic is primarily attributed to the sub-surface warm water intrusion from the Southern Ocean, the Larsen Ice shelf is more sensitive to surface atmosphere. This study further attributed the surface melting of the Larsen Ice shelf to the teleconnections from the (southern) tropical Pacific. Most of the statistical analysis and model experiments are

sound, although I do have some concerns on the mechanisms:

1) There doesn't seem to be a strong coherency between the anomalies of central Pacific SST and OLR associated with the Larsen melting (Fig. 1b, and the green box in Fig. 1d). This brings up an interesting question, as what causes the OLR anomaly (the convection)? Additionally, the trend of central Pacific convection appears vague (Fig. 1e) compared with other signals.

2) In this study, the authors mainly focus on the effect of a small area over the central Pacific (see the green box in Fig. 1d). However, according to the correlation map (Fig. 1b, d), the surface melting over the Larsen C glacier is significantly correlated with the SST over the tropical Southeast Pacific (positive) and with the South Pacific convergence zone (SPCZ) area (negative), and with the OLR over the central Pacific (negative, green-box in Fig. 1d, which is the main focus of this study), the tropical Southeast Pacific (positive), the western tropical Atlantic (negative) and the southern sub-tropical Indian Ocean (positive). Anomalies over the above areas may force the atmosphere circulation simultaneously. Considering that the correlation between the central Pacific convection and the Larsen ice shelf melting is not very high, singling it out from an extensive background with different signals may over estimate its impact. I suggest the authors also consider the other components of the tropical belt.

Reviewer #3 (Remarks to the Author):

The manuscript discusses the climatological factors influencing surface melting on the Larsen Ice Shelf, focussing on the Larsen C ice shelf because its controls are less well identified in existing literature. The authors show that, while increasing SAM trends have a lot to do with increased melting of the northernmost Larsen Ice Shelf components, for the Larsen C the key components are enhanced deep atmospheric convection in the central tropical Pacific and wave trains emanating from that region -- as well as a secondary influence related to the ENSO cycle. A critical steering component is high pressure centered in the Drake Passage, pushing airflow southeastward to the southern Larsen C and then northeastward on the lee side, creating strong foehn conditions and in some cases, record warm air temperatures along the eastern coast.

This is an excellent paper, and my comments are relatively minor - I have many edits and suggestions embedded in the .pdf I am submitting with this review.

In my experience (but not a quantitative analysis) melting in the -northern- Larsen ice shelf components (A and B) had a lot to do with a high pressure somewhat further east than show, with the center near the Malvinas / Falklands or the ocean area between then and Patagonia. just an observation, but steering the winds southeastward, directly across the peninsula (before they turn north) appears to be important for melting on both 'ends' of the Larsen.

I would suggest that a better title would include the term "Larsen C" - or some other way of indicating that this is not an analysis (another analysis) of the melting events that led to the well-researched break-ups in 1995 and 2002

Very clear, very convincing work.

Ted Scambos

We are very grateful for the three reviewers' careful reading of our manuscript and helpful comments. The points raised by the reviewers inspired us to carry out new investigations into *what triggers CPAC convection and is CPAC convection/Larsen C surface melt connected to AR activity on interannual timescales*, and have added the new results to the revised manuscript as new Figs. 5 and 6 and Extended Fig. 1. As detailed below, we feel the results from these new analyses have substantially strengthened the study by clarifying the physical mechanism triggering CPAC convection and its relationship with AR activity and Larsen C surface melt. These results have also broadened the study's implications by identifying mechanisms driving convection in the South Pacific Convergence Zone (SPCZ). We are confident these new analyses have addressed the major points made by the reviewers.

Firstly, we have now identified mid-latitude cyclonic activity to the south of CPAC and associated cold frontal intrusions into the tropics as the primary mechanism triggering CPAC convection. There is no significant connection between CPAC convection and local SST anomalies (i.e., the convection does not appear triggered by surface heating), although we do note enhanced CPAC convection is *broadly* associated with positive SST/convective anomalies across the central tropical Pacific, i.e., El Niño-like conditions, but given the stronger relationship with the local baroclinic zone along the cold front, the SST anomalies appear to play more of a conditional background role, likely favoring an unstable environment aiding in the development of intense convection when cold fronts arrive. This helps to explain the weak preference for El Niño conditions during the highest seasonal/monthly Larsen C surface melt years in Table 2 (i.e., nine of the 15 highest melt years occurred with El Niño), but importantly there is no significant correlation between surface melt and ENSO or equatorial Pacific SSTs on interannual timescales, and strong CPAC convection is noted during La Niña and ENSO-neutral condition. These results also imply the EPAC SST correlation we originally highlighted is likely a spurious correlation and is not a relevant physical driver of CPAC convection or the forced circulation pattern; in fact, it's possible positive EPAC SST anomalies could *be caused by the mid-latitude cyclone* that triggers the CPAC convection, and the associated northwesterly flow across the southeast tropical Pacific. We have thus removed this result and replaced the EPAC SST time series (Fig. 1c) with the SOI time series to highlight the lack of co-variability between surface melt and ENSO.

Secondly, we have identified a significant interannual relationship between Antarctic Peninsula AR activity and Larsen C surface melt ($r = 0.79$), and CPAC convection ($r = -0.70$). These results expand on our original analysis linking ARs to the two extreme AP temperature events, and now show CPAC convection is also an important driver of AR activity across the southwest AP, and AR activity is an important driver of Larsen C surface melt, on interannual and synoptic timescales.

Altogether, we feel these new results have significantly strengthened the study and we are grateful for these comments.

Please find below our point-by-point replies (in boldface) to all the reviewers' comments (in italics).

Reviewer #1 (Remarks to the Author):

This paper examines the extratropical response in the SH to tropical SST changes with an emphasis on extreme high temperatures and surface melt on the Larsen ice shelf, focusing on the influences of CP convection on atmospheric circulation around the AP. A series of sensitivity experiments are performed to examine this idea, and diagnostic analyses are used to show circulation and wave activity changes proposed to give rise to the response. The experiments and analyses are interesting, and the suggested dynamical mechanisms seem reasonable. The main comments below are aimed at helping to make the messages of the manuscript more concrete and accurate. It is very interesting that the two extreme AR events are connected with strong MJO bursts in the deep tropics on shorter time scales, which is consistent to the connections on interannual time scales. I think this study could be more useful and have more impact if the authors could do more to understand how AR activity is linked to the wave train pattern on interannual time scales. I believe the paper merits publication in Nature communications after minor revision.

We thank Reviewer #1 for their helpful and constructive comments. We have carried out new analyses to address points 2 and 3 below; point 2 is also raised by Reviewer #2.

1. Impacts of anthropogenic forcing and the stratospheric ozone depletion over the region are not discussed in the paper. I like to hear more thoughts from the authors about how to understand the role of external emission forcing (CO₂ and the ozone depletion) in determining this tropically-driven framework over the period. How do CMIP5 or 6 historical simulations replicate observed surface warming and ice shelf instability over the AP over the past decades (any studies have looked at this before)? Is this tropical-extratropical connection internally generated or externally driven? Without some additional thoughts on this aspect, I am not sure how to use what we learn from this study to guide us improve the future climate projection over the region?

This is an excellent question, but there are many challenges that limit the feasibility to perform this type of analysis, and moreover, we feel this question is outside the scope of our present study.

Firstly, the purpose of our study is to demonstrate the *internal dynamic* mechanism (during summer) leading to surface melt on Larsen C and the occurrence of extreme warm temperature events, and to contrast this with the well-known SAM mechanism that leads to warming on the northern AP. That is, CPAC convection *internally* forces an anomalous atmospheric circulation pattern (distinctly different from the SAM pattern) that creates favorable conditions for transporting extreme heat and moisture from low latitudes to the southwest AP, along with producing a significant increase in landfalling ARs (see our response to your point #3 below). In particular, the CPAC-forced zonally asymmetric circulation produces the strong anticyclone over Drake Passage (not a characteristic feature of SAM, or ENSO for that matter) that is crucial for rapidly transporting low-latitude heat and moisture poleward to the AP, and steering the flow southwesterly across the AP to produce strong surface melt on Larsen C. In response to your point #2 below, we have now identified the internal dynamic mechanism that triggers CPAC convection, and we don't find a significant relationship with any particular mode of tropical variability such as ENSO or local SST anomalies.

Furthermore, there are several problems with using CMIP models in the way suggested: (i) our records of temperature and melt on Larsen C are too short to allow a sensible comparison of observed trends and modelled historical runs. (ii) If instead of comparing long-term trends we compare year-to-year variations, we run into the problem that a free-running coupled climate model may not reproduce the year-to-year variations that we see in our single observed realization of the climate – i.e. we wouldn't necessarily expect the modelled CPAC OLR to be well-correlated with observed Larsen C temperature/melt. (iii) We might hope to find a correlation between modelled CPAC OLR and modelled Larsen C temperature/melt, but the AP orography is poorly represented in models of typical CMIP resolution, so foehn processes will be much weaker or non-existent compared to reality (for example, our CESM experiment reproduces the temperature/precipitation pattern (Figs. 4c,d) associated with strong foehn warming/melt on Larsen C from observations, but the simulation is too coarse to capture a local foehn warming signal on Larsen C).

Altogether, we are hopeful that our findings will help reconcile the *internal* mechanisms governing surface melt and extreme warming events on the AP, mechanisms that we expect will continue internally in the climate system into the future amidst externally forced climate change. We are also confident that these results will help to reconcile the impact of a continued positive SAM phase/trend on the remaining eastern AP ice shelves over the coming decades due to increasing greenhouse gases, as our study reveals the unique circulation pattern governing surface melt over the central/southern eastern AP and Larsen C ice shelf, while it is well known that positive SAM leads to warming on the northern AP. In particular, based on our results without the Drake Passage anticyclone and the southwesterly flow (neither of which is a feature of the summer SAM pattern), it is unlikely Larsen C will experience strong surface melt, and extreme high temperatures also appear tied to the Drake Passage anticyclone. We are confident these results will be helpful in reconciling future eastern AP ice shelf surface mass balance and stability through a improved understanding of the internal dynamic mechanisms responsible for this activity.

2. Is the topical mode identified here similar to the CP EL Nino or other SST modes, such as the IPO or PDO? More analysis could be done to shed more light on the origin of this tropical convection over the CP.

To investigate this question, we first revisited the 24 March 2015 and 6 February 2020 events to see what triggered the burst of CPAC convection. As shown below in Response Fig. 1 (which we have added as new Extended Figure 1 in the manuscript), we found in both events a sub-tropical (frontal) low developed south of CPAC several days prior to the event, with a pronounced surface cold front advancing northeast into the tropics preceded by an intense band of convection across the CPAC region. The convection persisted for around 3-5 days, during which it triggered the poleward wave fluxes (not shown). This scenario is consistent with previous studies identifying this region of the eastern SPCZ to exhibit high frequency variability that is attributed to interactions with mid-latitude wave activity (e.g., (Matthews 2012)).

22 March 2015

2 February 2020

Response Figure 1. (a-b) 925 hPa total temperature advection (shaded) and (c-d) OLR (shaded) anomalies alongside MSLP (hPa, contours) and 10m wind (ms^{-1} , vectors) anomalies preceding the (left) 24 March 2015 event and (right) 6 February 2020 event.

Two to four days prior to the events, a sub-tropical low (denoted by an “L”) develops south of CPAC with well-defined cold/warm advection regimes and a pronounced near-surface cold front (drawn as a curved black line) advancing northeastward into the tropics, which triggers an intense band of convection across CPAC (i.e., the $\sim 10\text{-}15^{\circ}\text{S}$, $165\text{-}150^{\circ}\text{W}$ region).

What about MJO? As shown in Response Fig. 2 below, the MJO was strongly active during the March 2015 event (as highlighted by previous work (Rondanelli et al. 2019)), but only weakly active during the February 2020 event. We infer from our synoptic analysis in Response Fig. 1 that the surface cyclone to the south of CPAC and its cold front were the primary physical mechanisms triggering the localized convection in CPAC during these two events, with MJO and perhaps SST anomalies playing a background role favoring an unstable environment. This conclusion is further justified in our next investigation of CPAC convection on interannual timescales.

Response Figure 2. The standardized pentad MJO indices from the Climate Prediction Center (www.cpc.ncep.noaa.gov/products/precip/CWlink/daily_mjo_index/pentad.shtml) across the 120°E (Index 3) to 40°W (Index 7) zones for the two weeks leading up to the (top) 24 March 2015 and (bottom) 6 February 2020 events; the gray shading denotes the CPAC region (10-15°S, 170-165°W). Positive values refer to a convectively “active” phase.

During March 2015 (top), the MJO was strongly active during and the week preceding the event (yellow and light blue lines), while the MJO was only weakly active during the February 2020 (bottom) event (gray and yellow lines). Indeed, the MJO, would favor ascent across the equatorial Pacific, but Response Fig. 1 shows the transient burst of convection in CPAC was generated south of the Equator along the cold front.

Next, in Response Fig. 3 below (which we have added as new Figure 5 in the manuscript) we investigated CPAC OLR interannual correlations and regressions with MSLP, low-level (925 hPa) thermal advection, and SST.

Response Figure 3. The detrended DJFM (left) CPAC OLR correlations and (right) standardized CPAC OLR regressions with (a-b) OLR, (c-d) MSLP, (e-f) 925 hPa temperature advection, and (g-h) SST during 1991-2015. Also shown on the right are regressions with 10m wind (for (b), (d), and (h)) and 925 hPa wind (for (f)). The correlations/regressions are multiplied by -1 to show conditions associated with enhanced CPAC convection. The bold contours denote correlations and regressions significant at $p < 0.10$, and wind vectors are shown only if at least one regression component is significant at $p < 0.10$. The CPAC area ($10-15^{\circ}S$, $170-165^{\circ}W$) is denoted by the black box, and the position of the surface low pressure center and its cold front are denoted by an “L” and curved black line, respectively.

CPAC OLR is broadly correlated with central tropical Pacific convection, but notably it is more strongly associated with an off-equatorial diagonal band of convection across the 10-15°S latitudes more characteristic of SPCZ convection. CPAC OLR is broadly correlated with low pressure across the east-central tropical Pacific, but consistent with the two case studies, the strongest negative MSLP correlations and regressions are located southeast of CPAC extending from around 25°S, 150°W poleward into mid-latitudes. The thermal advection and wind regressions clearly show this sub-tropical/mid-latitude cyclone produces southerly cold advection into the tropics with a well-defined cold front/baroclinic zone located directly over the CPAC region coinciding with the off-equatorial diagonal portion of the deep convection. Lastly, while enhanced CPAC convection is broadly associated with positive SST anomalies across the central tropical Pacific, we note the strongest SST correlations/regressions do not immediately overlap with the strongest OLR correlations/regressions, and there is a much closer alignment of the CPAC convection with the position of the cold front. Therefore, we conclude that the anomalous convection, particularly the off-equatorial diagonal portion characteristic of CPAC, is more tied to the low-level dynamic forcing rather than surface/SST heating anomalies.

3. Seasonal mean AR occurrence over the AP can be calculated based on the ERA5 AR dataset. If so, it is worthwhile to further calculate the connection between this AR occurrence and the circulation pattern (the high-pressure center around the AP) or the tropical mode on year to year time scales, as the authors have done in Fig. 2. The idea of the test is to examine how the overall activity of AR over the region throughout the season responds to large scale circulation forcing.

Thank you for this great suggestion. We have carried out new analyses of the relationship between AP AR activity and Drake Z500, CPAC OLR, and Larsen C surface melt, shown below in Response Fig. 4 (which we have added as new Figure 6 in the manuscript). Using the (Guan and Waliser 2015) AR catalog based on ERA-Interim, we calculated the total number of DJFM ARs that made landfall on the AP, and the total number of “extreme” landfalling ARs in which landfalling AR days had IVT values over the AP above the 95th percentile based on the DJFM IVT climatology. In addition, we examined changes in AR frequency during anomalous CPAC convection (days with daily CPAC OLR < -0.5 σ).

Interannual variability in the number of extreme landfalling ARs is very strongly correlated with Larsen C surface melt, Drake Z500, and CPAC OLR at 0.79, 0.64, and -0.70, respectively, indicating the number of landfalling ARs has a very important role in modulating seasonal total Larsen C surface melt (explaining over 60% of the interannual variability), and the number of landfalling ARs is strongly tied to CPAC OLR variability and the associated Drake Passage anticyclone.

The connection with CPAC OLR is similar on synoptic timescales, demonstrated by the significant increase (15-20%) in AR frequency across the Bellingshausen Sea and southwest AP during days of strong CPAC convection.

Response Figure 4. (a-c) Time series of the number of extreme landfalling ARs during DJFM (right y-axis) alongside the DJFM (a) Larsen C surface melt, (b) Drake Z500, and (c) CPAC OLR. Inset is the correlation coefficient of the two timeseries. (d) Composite anomaly of AR frequency for anomalous CPAC convection days (CPAC OLR $\leq -0.5\sigma$) showing the percentage difference relative to the DJFM AR frequency climatology. Stippling in (d) denotes anomalies significant at $p < 0.10$.

Reviewer #2 (Remarks to the Author):

This study attempts to build a linkage between the atmospheric deep convection over the central Pacific and the recently observed high-temperature over the Larsen ice shelves in austral summer. By using statistical analysis and numerical model simulations, the authors reveal that deep convection over the central Pacific may generate a stationary Rossby wave train, which further forms an anomalous high-pressure center west to the Peninsula, driving westerly wind anomalies, transporting heat and moisture from lower latitudes, and producing strong surface warming and melting.

Over all I think that this is a very interesting study, and may potentially be an important contribution to the field. While the accelerated ice shelf melting around West Antarctic is primarily attributed to the sub-surface warm water intrusion from the Southern Ocean, the Larsen Ice shelf is more sensitive to surface atmosphere. This study further attributed the surface melting of the Larsen Ice shelf to the teleconnections from the (southern) tropical Pacific. Most of the statistical analysis and model experiments are sound, although I do have some concerns

on the mechanisms:

1) There doesn't seem to be a strong coherency between the anomalies of central Pacific SST and OLR associated with the Larsen melting (Fig. 1b, and the green box in Fig. 1d). This brings up an interesting question, as what causes the OLR anomaly (the convection)? Additionally, the trend of central Pacific convection appears vague (Fig. 1e) compared with other signals.

Thanks for your comment, which was also raised by Reviewer #1 (please see above our reply to their 2nd point and our Response Figs. 1-3). We agree that while the Larsen C surface melt shows strong coherency with CPAC convection (i.e., detrended correlation of -0.63), CPAC convection did not show strong coherency with CPAC SST. This inspired us to carry out additional analyses to identify what causes the CPAC OLR anomaly. Our results, based on a synoptic analysis of the two record events and an interannual correlation and regression analysis, demonstrates CPAC convection is most strongly associated with a sub-tropical/mid-latitude cyclone located to the south of CPAC, and the CPAC convection, notably the off-equatorial diagonal structure, forms along the surface cold front/baroclinic zone. Local SST anomalies appear to play less of a role, and perhaps a background role in producing unstable environmental conditions that promote the development of deep convection, but our analysis indicates the cold front/baroclinic zone is the dominant mechanism, especially given the transient nature of CPAC convection which is unlikely to be caused by SST anomalies. This is consistent with previous studies showing this eastern region of the SPCZ to exhibit high frequency variability tied to mid-latitude wave activity (e.g., (Matthews 2012)).

2) In this study, the authors mainly focus on the effect of a small area over the central Pacific (see the green box in Fig. 1d). However, according to the correlation map (Fig. 1b, d), the surface melting over the Larsen C glacier is significantly correlated with the SST over the tropical Southeast Pacific (positive) and with the South Pacific convergence zone (SPCZ) area (negative), and with the OLR over the central Pacific (negative, green-box in Fig. 1d, which is the main focus of this study), the tropical Southeast Pacific (positive), the western tropical Atlantic (negative) and the southern sub-tropical Indian Ocean (positive). Anomalies over the above areas may force the atmosphere circulation simultaneously. Considering that the correlation between the central Pacific convection and the Larsen ice shelf melting is not very high, singling it out from an extensive background with different signals may over estimate its impact. I suggest the authors also consider the other components of the tropical belt.

While we agree other tropical regions may contribute to the circulation pattern identified in our study, we feel the results of our three separate and independent investigations support the conclusion that CPAC convection is the dominant mechanism triggering the circulation pattern. Specifically:

- 1. The Larsen C surface melt/CPAC OLR correlation is very strong at -0.63 (detrended) and is the highest OLR correlation of any region in the tropics (Fig. 1d);**
- 2. The spatial pattern of the Larsen C surface melt (Fig. 1f,h) and the CPAC OLR (Fig. 2) correlations with upper-tropospheric streamfunction and Z500 are very similar, and both show a wave train emanating from the anomalous upper tropospheric anticyclone/mass convergence on the poleward edge of CPAC convection;**

3. The two synoptic case studies objectively show the wave train that caused the record temperature events (which had a strong Drake Passage anticyclone and an AR) are both tied to anomalous poleward wave fluxes emanating from the CPAC convective anomaly;
4. Lastly, and arguably most importantly, the sensitivity experiment we carried out isolated the direct influence of CPAC on the atmosphere and removed effects of all other tropical regions by setting their SSTs to climatology. The isolated direct influence of CPAC reproduces the circulation pattern tied to Larsen C surface melt remarkably well, in particular the elongated mid-latitude cyclone and the Drake Passage anticyclone, and the warm/moist air convergence across the southwest Peninsula.

For these reasons, we feel our conclusions are sound and based on a comprehensive suite of analyses, and the role of the rest of the tropics is outside the scope of our study.

Reviewer #3 (Remarks to the Author):

The manuscript discusses the climatological factors influencing surface melting on the Larsen Ice Shelf, focussing on the Larsen C ice shelf because its controls are less well identified in existing literature. The authors show that, while increasing SAM trends have a lot to do with increased melting of the northernmost Larsen Ice Shelf components, for the Larsen C the key components are enhanced deep atmospheric convection in the central tropical Pacific and wave trains emanating from that region -- as well as a secondary influence related to the ENSO cycle. A critical steering component is high pressure centered in the Drake Passage, pushing airflow southeastward to the southern Larsen C and then northeastward on the lee side, creating strong foehn conditions and in some cases, record warm air temperatures along the eastern coast.

This is an excellent paper, and my comments are relatively minor - I have many edits and suggestions embedded in the .pdf I am submitting with this review.

Thank you very much for your positive and supportive comments, and for the very helpful edits and suggestions which have helped clarify our results and implication. We have replied point by point to your comments in the separately attached PDF.

In my experience (but not a quantitative analysis) melting in the -northern- Larsen ice shelf components (A and B) had a lot to do with a high pressure somewhat further east than show, with the center near the Malvinas / Falklands or the ocean area between then and Patagonia. just an observation, but steering the winds southeastward, directly across the peninsula (before they turn north) appears to be important for melting on both 'ends' of the Larsen.

I would suggest that a better title would include the term "Larsen C" - or some other way of indicating that this is not an analysis (another analysis) of the melting events that led to the well-researched break-ups in 1995 and 2002

Excellent point. Indeed, our work focuses on Larsen C and does not specifically examine the Larsen A or B region. We have revised the title of the manuscript to specify Larsen "C".

Very clear, very convincing work.

Thank you so much for your support of our work!

Ted Scambos

References

Guan, B., and D. E. Waliser, 2015: Detection of atmospheric rivers: Evaluation and application of an algorithm for global studies. *J. Geophys. Res. Atmospheres*, 120, 12514–12535, <https://doi.org/10.1002/2015JD024257>.

Matthews, A. J., 2012: A multiscale framework for the origin and variability of the South Pacific Convergence Zone. *Q. J. R. Meteorol. Soc.*, 138, 1165–1178, <https://doi.org/10.1002/qj.1870>.

Rondanelli, R., B. Hatchett, J. Rutllant, D. Bozkurt, and R. Garreaud, 2019: Strongest MJO on Record Triggers Extreme Atacama Rainfall and Warmth in Antarctica. *Geophys. Res. Lett.*, 46, 3482–3491, <https://doi.org/10.1029/2018GL081475>.

REVIEWER COMMENTS

Reviewer #1 (Remarks to the Author):

I am happy with what the authors did to address the concerns i raised last time. So, I am confident that the paper is greatly improved and should be accepted as it is now.

Reviewer #2 (Remarks to the Author):

I thank the authors for their hard work in responding to my original comments. The manuscript has been improved, although I still have some concerns about the second comment in my previous review.

"2) In this study, the authors mainly focus on the effect of a small area over the central Pacific (see the green box in Fig. 1d). However, according to the correlation map (Fig. 1b, d), the surface melting over the Larsen C glacier is significantly correlated with the SST over the tropical Southeast Pacific (positive) and with the South Pacific convergence zone (SPCZ) area (negative), and with the OLR over the central Pacific (negative, green-box in Fig. 1d, which is the main focus of this study), the tropical Southeast Pacific (positive), the western tropical Atlantic (negative) and the southern sub-tropical Indian Ocean (positive). Anomalies over the above areas may force the atmosphere circulation simultaneously. Considering that the correlation between the central Pacific convection and the Larsen ice shelf melting is not very high, singling it out from an extensive background with different signals may over estimate its impact. I suggest the authors also consider the other components of the tropical belt."

I agree that all evidences listed in the authors response may indicate that the central Pacific contributes to the Larsen Ice Shelf warming. However, according to Figure 1d (and 1b), the eastern Pacific and the western Atlantic can be at least equally important. I still suggest the authors clarify the respective contributions by the variability over these two regions, especially considering that the title of the paper is "Central tropical Pacific convection drives extreme high temperatures ..."

Reviewer #1 (Remarks to the Author):

I am happy with what the authors did to address the concerns i raised last time. So, I am confident that the paper is greatly improved and should be accepted as it is now.

We thank the reviewer for all their helpful comments on the previous draft of our manuscript and for their positive recommendation of our study.

Reviewer #2 (Remarks to the Author):

I thank the authors for their hard work in responding to my original comments. The manuscript has been improved, although I still have some concerns about the second comment in my previous review.

“2) In this study, the authors mainly focus on the effect of a small area over the central Pacific (see the green box in Fig. 1d). However, according to the correlation map (Fig. 1b, d), the surface melting over the Larsen C glacier is significantly correlated with the SST over the tropical Southeast Pacific (positive) and with the South Pacific convergence zone (SPCZ) area (negative), and with the OLR over the central Pacific (negative, green-box in Fig. 1d, which is the main focus of this study), the tropical Southeast Pacific (positive), the western tropical Atlantic (negative) and the southern sub-tropical Indian Ocean (positive). Anomalies over the above areas may force the atmosphere circulation simultaneously. Considering that the correlation between the central Pacific convection and the Larsen ice shelf melting is not very high, singling it out from an extensive background with different signals may over estimate its impact. I suggest the authors also consider the other components of the tropical belt.”

I agree that all evidences listed in the authors response may indicate that the central Pacific contributes to the Larsen Ice Shelf warming. However, according to Figure 1d (and 1b), the eastern Pacific and the western Atlantic can be at least equally important. I still suggest the authors clarify the respective contributions by the variability over these two regions, especially considering that the title of the paper is “Central tropical Pacific convection drives extreme high temperatures ...

We thank the reviewer for their suggestion to consider the potential role of other tropical regions. We have re-examined the SST and OLR correlation maps and considered the potential for Rossby wave development in the other areas of significant correlation. We have carried out a new analysis of climatological omega (known to govern updraft formation needed for Rossby wave development, e.g., Lachlan-Cope and Connolley 2006) and reviewed the results of previous studies and numerical experiments that have specifically examined the Southern Hemisphere circulation response to forcing from other tropical regions, in particular the tropical Atlantic. Based on the results of these new analyses, we are even more confident that central tropical Pacific convection is the dominant mechanism forcing the anomalous atmospheric circulation pattern that leads to surface melt on Larsen C. We have added a new discussion to the manuscript, detailing the potential role of other tropical regions contributing to Larsen C surface melt, along with the relevant citations, and we have added the Response Fig. 1 below to the manuscript as new Extended Fig. 1.

Response Figure 1. The DJFM detrended correlation (shaded) and $p < 0.10$ significance (bold contours) of Larsen C surface melt with (a) SST and (b) OLR, as in Figs. 1b and 1d, respectively, along with the 1979-2019 ERA5 climatological omega at 600 hPa (thin dashed contours; hPa second⁻¹). Only negative omega values (areas of climatological ascent) are shown, which denotes regions that are favorable for the development of deep tropical convection and upper-tropospheric divergent flow required to generate a Rossby wave, as detailed below.

CPAC is the only region where a significant correlation with Larsen C surface melt aligns with environmental conditions that are physically conducive for deep convection, divergent forcing, and Rossby wave development.

1. The positive SST correlation in the southeast Pacific lies in an area of strong climatological subsidence. SST anomalies in this region are not able to generate deep convection and upper-tropospheric divergent forcing needed to produce a Rossby wave (Lachlan-Cope and Connolley (2006) (see below for more details).
2. The significant negative SST correlations east of New Zealand also lie in a region of climatological subsidence. As we previously stated, these SST anomalies are likely tied to the local synoptic pattern that triggers the CPAC convection, i.e. the cold southerly surface winds west of the sub-tropical cyclone (Fig. 5 in our manuscript).
3. While there is a small region of significant positive SST correlations north of Australia, there is no significant correlation with OLR in this region, and without a convective anomaly associated these SST anomalies, it is unlikely this region makes a major contribution to the forced circulation pattern associated with surface melt.
4. The only other significant correlation is in the western tropical Atlantic, which shows a positive OLR anomaly (and weak negative SST anomalies) is associated

with surface melt. This anomaly also lies in an area of climatological subsidence, but moreover, as discussed below, extensive research has already been carried out examining the tropical Atlantic teleconnection to the Southern Hemisphere (Li et al. 2014, 2015; Simpkins et al. 2014), and the anomalous circulation that would be expected from these SST/OLR anomalies is inconsistent, opposite in fact, the circulation pattern associated with enhanced surface melt. Therefore these anomalies are also highly unlikely to contribute to the surface melt, and if anything, this forcing would dampen the circulation anomalies forced by CPAC.

In summary, the only region of significant SST/OLR correlation that aligns with environmental conditions conducive for deep convection and Rossby wave development is in CPAC. In fact, CPAC lies in one of the most favorable regions for deep convection throughout the entire tropics, and therefore this region would be very conducive for strong ascent and divergent forcing to produce a Rossby wave, further supporting our conclusion.

Figure 4. Omega anomalies at 500 hPa for a +2°C SST anomaly in the mid-Pacific (anomaly marked with square).

Lachlan-Cope and Connolley (2006)-JGR

Figure 4 (left) from Lachlan-Cope and Connolley (2006) shows the anomalous ascent associated with a +2°C surface heating anomaly over a large 40 x 30° box in the southeast Pacific. The positive SST anomaly only produces an increased updraft (needed for Rossby wave development) in the area of climatological ascent north of the Equator, with virtually no change in the southeast tropical Pacific which is an area of climatological subsidence. (Model: HadAM3).

Figure 5. Omega anomalies at 500 hPa plotted against SST anomaly for a series of SST anomalies positioned in the mid-Pacific (15°S 105°W).

Lachlan-Cope and Connolley (2006)-JGR

Figure 5 (right) from Lachlan-Cope and Connolley (2006) specifically examines how atmospheric ascent responds to increasing SST forcing over the southeast tropical Pacific region where the significant positive correlation with surface melt is seen (15°S, 105°W). There is no updraft response to positive SST anomalies in the southeast Pacific for any realistic SST anomaly below +4°C. Only until the SST anomaly exceeds +4°C does an updraft develop and strengthen, which is unrealistic and larger than any ENSO SST anomaly ever observed. (Model: HadAM3)

Li et al. 2014-Nature

Figure 2g from Li et al. (2014) shows the atmospheric response to observed monthly SST variability in the tropical Atlantic (20°N–20°S) that comprises a strong warming trend. *SST warming in the tropical Atlantic* is associated with a large deep cyclone in the South Pacific and warming on the Antarctic Peninsula. In contrast, the correlations with Larsen C surface melt suggest a relationship with *cold* SST anomalies in the tropical Atlantic and a positive OLR anomaly/reduced convection. Therefore, based on these previous results, tropical Atlantic SST cooling would be expected to cause an *anticyclone* over the South Pacific, which would *cool* the Peninsula and is inconsistent, and nearly opposite, the circulation pattern that is associated with surface melt. (Model: CAM4)

[Redacted]

More specifically for summer (DJF), Figure 3f from Li et al. (2015) shows the atmospheric response to observed *DJF SST warming in the tropical Atlantic (20°N-20°S)*, and so again the melt correlations (SST cooling) would be associated with circulation anomalies opposite to those shown in Fig. 3f. DJF SST cooling in the tropical Atlantic would be associated with an *anticyclone over the Ross Sea* and a *cyclone near Drake Passage*. This is again opposite the circulation pattern that results in enhanced Larsen C surface melt and would be associated with cooling on the Peninsula. (Model: CAM4)

[Redacted]

Lastly, the results of Li et al. (2014, 2015) using CAM4 are further corroborated in Fig. 3c above from Simpkins et al. (2014) using CAM3. SST warming in the tropical Atlantic produces a large deep cyclone in the South Pacific. Therefore, SST cooling/reduced convection in the tropical Atlantic would be associated with an anticyclone over the South Pacific and cooling on the Peninsula. (Model: CAM3)

Conclusions

- ***Positive SST anomalies in the southeast tropical Pacific lie in a region of strong climatological subsidence, and they would not be physically capable of producing the ascent required for Rossby wave development and a high-latitude teleconnection. Therefore, as we discussed in our previous response letter, the significant positive SST correlation in the southeast tropical Pacific is likely a spurious correlation associated with the local synoptic pattern that triggers CPAC convection (i.e., the northwesterly surface winds in the warm sector of the synoptic cyclone).***
- ***Similarly, negative SST anomalies south of the SPCZ/east of New Zealand also lie in a region of climatological subsidence, and they too would not be favorable sources of ascent strong enough to trigger a Rossby wave, and the significant negative correlations here are also likely a spurious correlation associated with the synoptic pattern that triggers CPAC convection (i.e., the cold southerly winds behind the cold front). This is also consistent with Clem et al. (2019)-GRL examining the SPCZ teleconnection associated with SST anomalies along the southern edge of the SPCZ, and a convective response to SST heating only occurs within the narrow region of climatological ascent in the SPCZ, with the convective response rapidly diminishes south of this region.***

- Apart from a small region of positive SST correlations north of Australia where no OLR correlation exists, the only other significant correlation is a positive OLR correlation in the western tropical Atlantic that is also associated with weak negative SST anomalies. Multiple studies have examined the Southern Hemisphere atmospheric circulation response to tropical Atlantic variability, and warming/increased convection in the tropical Atlantic is associated with a deep cyclone in the South Pacific and an anticyclone over Drake Passage resulting in warming on the Antarctic Peninsula. Therefore, the SST/OLR anomalies reflected in the correlations would be associated with an *anticyclone in the South Pacific* and a *cyclone over Drake Passage* that would lead to *cooling* on the Peninsula. Therefore, it is very unlikely the tropical Atlantic correlations reflect a physical connection to Larsen C surface melt.
- CPAC lies in an extremely favorable region for deep ascent and strong divergent forcing that can trigger a Rossby wave. And indeed, we find clear evidence from three independent analyses that point to the dominant role of CPAC convection in forcing the circulation pattern that governs interannual variability in Larsen C surface melt, atmospheric river activity, and the occurrence of extreme events.

We are grateful to the reviewer for making these suggestions, as these new results give us increased confidence in the dominant role of CPAC. Further, we have added these results and this discussion to the manuscript to further clarify and support our conclusions.

References

Clem, K. R., B. R. Lintner, A. J. Broccoli, and J. R. Miller, 2019: Role of the South Pacific Convergence Zone in West Antarctic Decadal Climate Variability. *Geophys. Res. Lett.*, 46, 6900–6909, <https://doi.org/10.1029/2019GL082108>.

Lachlan-Cope, T., and W. Connolley, 2006: Teleconnections between the tropical Pacific and the Amundsen-Bellinghausens Sea: Role of the El Niño/Southern Oscillation. *J. Geophys. Res. Atmospheres*, 111, <https://doi.org/10.1029/2005JD006386>.

Li, X., D. M. Holland, E. P. Gerber, and C. Yoo, 2014: Impacts of the north and tropical Atlantic Ocean on the Antarctic Peninsula and sea ice. *Nature*, 505, 538–542, <https://doi.org/10.1038/nature12945>.

Li, X., E. P. Gerber, D. M. Holland, and C. Yoo, 2015: A Rossby Wave Bridge from the Tropical Atlantic to West Antarctica. *J. Clim.*, 28, 2256–2273, <https://doi.org/10.1175/JCLI-D-14-00450.1>.

Simpkins, G. R., S. McGregor, A. S. Taschetto, L. M. Ciasto, and M. H. England, 2014: Tropical Connections to Climatic Change in the Extratropical Southern Hemisphere: The Role of Atlantic SST Trends. *J. Clim.*, 27, 4923–4936, <https://doi.org/10.1175/JCLI-D-13-00615.1>.